# The impact of bite force on the stability of dental implants

**Nawres Bahaa Mohammed**[1], **Zina Ali Daily**[2], **Mohammed Rhael Ali**[3], **Noor Fathi Kazim**[4], **Hashim Mueen Hussein**[5]\*, **Athraa Ali Mahmood**[6]

**1** Department of Maxillofacial Surgery, College of Dentistry, Al-Ameed University, Karbala, Iraq, **2** Department of Periodontics, College of Dentistry, Al-Ameed University, Karbala, Iraq, **3** Department of Maxillofacial Surgery, College of Dentistry, Tikrit University, Tikrit, Iraq, **4** Department of Oral and Maxillofacial Radiology, Al-Mustaqbal University College, Babel, Iraq, **5** Department of Conservative Dentistry, College of Dentistry, Mustansiriyah University, Baghdad, Iraq, **6** Department of Periodontics, College of Dentistry, Mustansiriyah University, Baghdad, Iraq

\* hashimmueenhussein@gmail.com, hashimmueenhussein@uomustansiriyah.edu.iq

## Abstract

### Background

Since dental implants (DI) are endo-osseous implants that are inserted into the bone and periodontium, which ultimately support the occlusal load, a patient's bite force (BF) may overburden the supporting DI, leading to bone loss and DI failure. The purposes of this study were to explore the effect of the BF on DIs in different locations of jaws, of both genders during three visits, to investigate DI stability in different locations of jaws, of both genders during three visits, and to assess the association of BF and DI stability(ISQ) with the influence of variables (time, implant location, and gender).

### Methods

The current cohort study involved 80 individuals of both genders who had lost some teeth and needed DIs for the anterior and posterior regions of the jaws. After the insertion of the DIs, their BF was examined by the Loadstar™ sensor, and the DI stability was monitored at three visits of these individuals. First visit during the day of insertion of the crown on the abutment of the implant (immediately loading), second visit at six months following insertion, and third visit after 18 months following insertion.

### Results

The data on the anterior and posterior bite forces (BFs) for the male and female participants after the insertion of the DIs were analysed at three visits. On average, the males exhibited a significantly higher posterior BF compared to the females at the respective three visits. Similarly, the average posterior ISQ in the males was

---

**Data availability statement:** The data that support the findings of this study are available in the Supporting Information files. This project contains the following underlying data the data contains the recording of bite force of the male and female participants after the insertion of dental implants for the anterior and posterior regions of the jaws. The dental implants' stability was monitored at the anterior and posterior regions of both genders' jaws. For determining the impact of the BF on the stability of dental implants. Data are available under the terms CC0. All participants entered the study after they received full information about the nature, aims, processes of the study, data sharing, the anonymization of participants in deposited data, and publication of raw data (without containing the name of participants) before signing an informed written consent form.

**Funding:** The author(s) received no specific funding for this work.

**Competing interests:** The authors have declared that no competing interests exist.

significantly increased than in the females at the respective three visits. Results of the two-way ANOVA of all groups for the implant BF and ISQ values were significantly influenced by the interaction of time, locations of the implants, and gender. The association between the BF and DI stability was significant in using the difference in the regression coefficient ($b$), which was impacted by time, implant location, and gender.

## Conclusions

This study found the significantly complex interaction of factors (time, location, gender) influencing on change of BF and ISQ, by affecting the process of osseointegration. The significantly higher BF in the posterior regions of males, and significantly greater stability of DIs, could have a potential role in the best DIs therapy. In addition, the association of the BF and DI stability is significantly established with the most important factors influencing the change in BF and DI stability. It varies in a dynamic manner as the interface between the bone and the implant matures, and the patient's gender, time, and anatomical location all play a significant role in its context. The DIs that are loaded early is dependent on the BF. The BF may be crucial in determining the best DI stability.

## Introduction

Dental implants (DIs) have become a popular and effective option for replacing missing teeth [1,2]. One of the most important factors that affects the success of DIs is the occlusal force (OF) or the force that is applied to the DI by opposing teeth during biting and chewing [3,4]. The bite force (BF) can have both positive and negative effects on the DI, depending on the amount and direction of the force [5,6]. Early loading of a DI, which involves placing the DI rapidly after surgery, is significantly popular in the field of DI dentistry [7]. Several studies indicate that initiating early loading with a carefully regulated OF can improve the process of osseointegration and result in favourable results for the DI [8–10]. Nevertheless, an excessive or unregulated BF can lead to the failure of the DI, bone resorption, and fracture of the DI. The impact of the BF on an early-loaded DI can be influenced by several factors, including the DI design, DI site, bone quality, and the nature and intensity of the BF [11]. Dental implants (DIs) inserted into soft bones are more prone to injury due to the BF compared to those inserted into thick bones [12,13]. Applying a regulated amount of pressure on the teeth can improve the integration of the DI with the surrounding bone and increase the chances of a good outcome. However, an excessive or unregulated force might cause the DI to fail and lead to bone loss [14].

Bite force (BF) is the pressure applied by the jaws while chewing and biting. It is impacted by variables such as the quantity and arrangement of the teeth, the force exerted by the muscles involved, and the general oral well-being of an individual. An optimal BF is essential for effective chewing and ensuring the

durability of dental restorations, such as DIs. It differs from person to person and greatly impacts the DI system [15,16]. The DI abutment transmits the BF to the DI, which in turn transfers it to the crown denture or other prosthetic parts. It is crucial to assess and take into consideration the BF when planning and creating DI-supported restorations [17].

The BF is determined by several factors, including the characteristics relating to the patient and the prosthesis. The factors that are dependent on the patient are the patient's age, gender, occlusal habits, parafunctional behaviours, such as teeth grinding, and overall health of the stomatognathic system [18–20], while the prosthesis-related factors include the design, material, occlusal arrangement, and number of teeth supported by the DI [21]. A restored DI transmits multidirectional forces that alter the amount of axial, nonaxial, and transversal loads throughout the chewing process and movements of the jaw, affecting the connection between the DI and the bone [22]. Modifications to the design of the neck of the DI, which give rise to bone reabsorption, often take place in this area of highly concentrated mechanical forces [23]. The changes made to the neck of the DI are intended to lessen pressures like the forces of tension and shear in the cortical area [24,25].

While DIs offer numerous benefits, such as improved aesthetics and functionality [26,27]. The correlation of the BF with the DI stability is undetected until now. The research question is the BF associated with the stability of DIs. This study aims to explore the effect of the BF on DIs in different locations of jaws, of both genders, during three visits, to investigate DI stability in different locations of jaws, of both genders, during three visits, and to assess the association of BF and DI with the influence of variables (time, implant location, and gender). The hypothesis is that there was a change in BF and ISQ of DIs during 18 months, followed insertion of the crown on the abutment of the implant. The null hypothesis is that there was no change in BF and ISQ of DIs during 18 months, following the insertion of the crown on the abutment of the implant.

## Materials and methods

Study design, this cohort study used a prospective approach to analyse the clinical information and records of a group of patients who needed DI placements between February 2021 to June 2024. All procedures performed in this study involving human participants were in accordance with the Declaration of Helsinki and its later amendments for human research. The study was conducted in accordance with and approved by the Ethics Committee of the College of Dentistry, University of Al-Ameed, Iraq (#52017, 20 February/2021). Written informed consent was obtained from all subjects and/or their legal guardian (s), where all adult participants entered the study after they received full information about the nature, aims, processes of the study, data sharing, the anonymization of participants in deposited data and publication of raw data (without containing the name of participants) before signing an informed written Consent form. The current study followed the Strengthening the Reporting of Observational Studies in Epidemiology (STROBE) guidelines in terms of the study design and reporting of results.

### Sample size power analysis

The sample size, calculated by G power, comprised 18 individuals in each group at a power of 80 and an α probability of 0.05 that it depended on the pilot study resulting of 12 patients with DIs placed, three in each group (3 patients placed DIs for anterior regions of male, 3 patients placed DIs for posterior regions of male, 3 patients placed DIs for anterior regions of female and 3 patients placed DIs for posterior regions of female) that BF and ISQ measured during three visited interval (first visit during the day of insertion of the crown on the abutment of the implant (immediately loading), second visit at six months following insertion, and third visit after 18 months following insertion. Then the data was analyzed and resulting in a significant effect size of F at 0.4, and four groups. The total number of participants was around 80 to avoid patient dropouts, and these were divided into four groups. 80 individuals of both genders who had lost some teeth and needed DIs for the anterior and posterior regions of the jaws.

## Study population

The study included 80 patients at the Maxillofacial Surgery Department of the Dentistry College at the University of Al-Ameed who needed DIs during the study period, with three follow-up visits. First visit during the day of insertion of the crown on the abutment of the implant (immediately loading), second visit at six months following insertion, and third visit after 18 months following insertion. The inclusion criteria were male and female patients aged 25–40 years, who had sufficient bone volume for DI placements instead missing number of natural teeth and had natural occlusions, opposed natural teeth should present to implants that needed inserting, understood and signed an informed consent form, and were followed up at postoperative visits. Patients with a history of systemic diseases affecting bone metabolism, periodontitis, cigarette smoking, or drinking alcohol were excluded from the study. Moreover, the exclusion criteria included: they had missing permanent teeth which replaced by an implant, and a restoration with a crown of opposing occlusion, delayed loading, and parafunctional habits.

## Data collection

State data structure, subjects measured at 3 time points; BF and ISQ measured at the anterior/posterior area; gender as a between-subjects factor. Relevant clinical information, such as the patient's characteristics after placement with a DI by a specialist, medical background, X-ray pictures, and scans of the inside of the mouth, was obtained from the computerised medical records and documents of the patient. A Loadstar™ sensor was used to collect the BF values of posterior and anterior implant areas [28]. This sensor offers various force-measuring solutions with updated data rates of up to 50 KHz, making it suitable for applications that require the BF to be noted soon after the operation and during consultation intervals. Regular monitoring of the BF and DI stability was crucial in the post-implantation phase. Loadstar™ sensor records were utilised to assess the OF and identify any abnormalities or imbalances. While Osstell Implant Stability Quotient (ISQ) technology was used to evaluate the DI stability by examining the resonance frequency of the DI. The patients were also educated on proper oral hygiene practices, including avoiding excessive BF on the DI, maintaining regular dental visits, and promptly addressing any signs of discomfort or changes in the bite. The data availability was presented in supporting information files.

## Dental Implant (DI) characteristics

Information on the DIs of the research population, which were specifically manufactured by Straumann and Medintika (Straumann® Dental Implant System, USA, Medintika® Dental Implant, Germany), was carefully recorded, including their size (4.0*10, 3.5*11, 3.3*10, or 3.8*9), surface features, and prosthetic elements. Any alterations to the shape of the DI by modifications to the design of the neck of the DI or the process of surgery for improving the spread of the load and enhancing the stability were observed.

## Surgical procedure

The implants distributed were placed in the anterior and posterior regions of the jaws (Table 1). Following dental cone-beam computed tomography (CBCT), the quantity and quality of bone were assessed, and a stent was created for implantation at the proper location for each patient. No bone augmentation treatment was used throughout the implant placement process. As instructed by the manufacturer, the location was drilled using a point Lindemann drill first, then surgical drills. A skilled researcher meticulously drilled every dental implant bed at a consistent length and angles in order to produce comparable insertion torque values (ITV) of about 35 Ncm amongst the implants. A drilling machine specifically made for implant surgery was used for measuring ITV as much as 35 Ncm at around 20 rpm and 8 Hz. The drill unit's handpiece was the only tool used to put each implant. Following surgery, CBCT was utilized to assess the integrity of the osseous tissue surrounding the implant [28].

**Table 1. Demographic data.**

| Variables in groups | DIs for the anterior regions of males | DIs for the posterior regions of males | DIs for the anterior regions of females | DIs for the posterior regions of females | p value |
|---|---|---|---|---|---|
| Number of patients (total) | 20 | 20 | 20 | 20 | 0.11 |
| Age | 35 | 37 | 36 | 34 | 0.18 |
| Gender | 20 | 20 | 20 | 20 | 0.21 |
| Number of implants (total) | 45 | 76 | 43 | 71 | 0.79 |
| Implant diameter | | | | | |
| 3.3 mm | 8 | 10 | 7 | 12 | 0.122 |
| 3.5 mm | 19 | 22 | 23 | 21 | 0.084 |
| 3.8 mm | 14 | 24 | 11 | 30 | 0.056 |
| 4 mm | 4 | 20 | 2 | 8 | 0.077 |
| Length | | | | | |
| 9 mm | 13 | 49 | 15 | 42 | 0.095 |
| 10 mm | 25 | 22 | 19 | 23 | 0.082 |
| 11 mm | 7 | 5 | 9 | 6 | 0.112 |

Significant (*p*-values ≤ .05), non-significant (*p*-values > 05).

## Bite Force (BF) assessment

Bite force (BF) assessments were acquired utilising a Loadstar™ sensor (Loadstar sensor, DI-100U,16-bit load cell interface, Fremont, California) (Fig 1), which offers various force measurement devices with updated data rates of up to 50 KHz. This capability might be advantageous for applications such as failure-strength testing or material char-acterisation. The BF values were tested within the range of 100–400 N at particular time intervals. This systematic method was observed to guarantee standardization and dependability in the assessment of the BF. The protocol of maximum bite force estimation. The individual was told to sit on a dental seat without a head support, erect, comfort-able, and unstrained. A Loadstar sensor was used for determining the maximum force of biting (MBF) and vertically inter-occlusal BF on both sides. A USB cable is used to link the sensor to the PC. The first molars region (right and left) was where the load sensor was placed horizontally. Participants were instructed to bite as heavily as they could onto the load sensor for a brief period of time. A new record—the greatest—was established every second. Ten mea-surements of the recording table were selected. This method was performed three times on both sides, separated by two minutes, and the average result for every side was obtained as MBF. We calculated the maximal BF in Newtons [29]. The reliability of measurements was estimated by intra-operator evaluation and repeated test with the method of MBF.

## Stability measurements

The method for monitoring implant stability during healing and immediately loading is based on resonance frequency analysis (RFA) applied to the implant–bone interface. Devices using this method contain a transducer peg, which is con-nected to the implant and excited by magnetic waves over a range of frequencies. The frequency of the resultant vibration is automatically translated into an index called the implant stability quotient (ISQ), with values ranging between 0 and 100. The RFA values are a measure of the deflection of the implant–bone complex by the lateral forces applied by the trans-ducer and reflect the multidirectional fixation strength. The stability of each implant (one measurement from each of the 3 different directions) was measured with the Osstell (Integration Diagnostics AB, Göteborg, Sweden). The Osstell system is a magnetic detection device. RFA devices after the transducer (Smartpeg) was screwed to the implant to obtain the ISQ. The protocol of measurement of DI stability, an experienced, right-handed investigator, assessed the ISQ. ISQ was

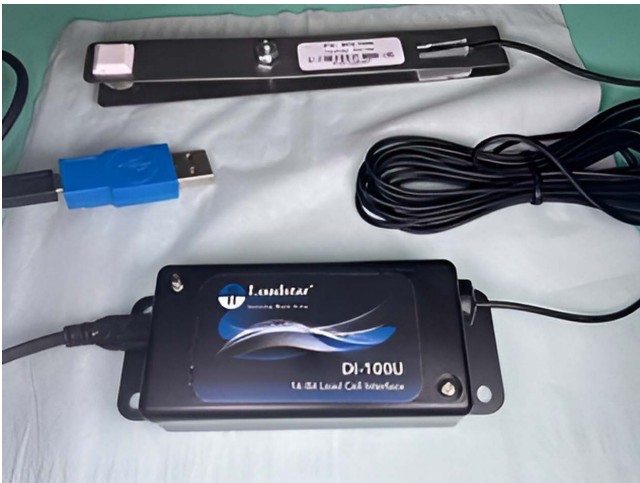

**Fig 1. Load star sensor.**

assessed using the Osstell in order to keep the fixing force of the implant from altering when the healing abutment was being installed and released. The smart peg was manually attached to the implant fixture in order to use the Osstell ISQ Mentor for evaluation. Every equipment was operated in compliance with the guidelines provided by the manufacturer. The Osstell ISQ mentor's manufacturer advises holding the gadget tip at a 45-degree angle and close (2.0–4.0 mm) to the smart peg surface, avoiding contact with it. The implants in each location were evaluated one after the other since testing a single implant twice could increase the precision of measurements. After the measurements with the Osstell ISQ Mentor, healing abutments were connected to the implant with 30 Ncm torque, using a torque ratchet, by selecting a height that could expose about 2.0 mm from the gingival level [30].

## Outcome measures

The main objective of the study was to determine the relationship between the BF and the stability of DIs during the 18 months that followed. The secondary outcomes included the influence of time, gender, and the DI location in the jaws on the BF and the DI stability.

## Statistical analyses

Descriptive statistics through GraphPad® Prism 9.5.1 were used to summarize the patient demographics, BF measurements, and DI stability measurements. The normality of distribution (parametric) was significantly result by used the Shapiro–Wilk. The parametric data was analyzed. Then the continuous variables were expressed as the mean standard deviation (±), while the categorical variables were presented as frequencies and percentages. Comparative analyses, such as an analysis of variance (One-Way Repeated-Measures ANOVA), were performed to assess differences in the BF and DI stability in males and females at the anterior and posterior areas of the jaws among three postoperative time points. The two-way and three-way ANOVA of all groups for the implant BF and ISQ values between the time, locations of the implants, and gender. The sphericity (Mauchly) and handling of violations (Greenhouse-Geisser) were employed. For comparisons between more than two groups, the Bonferroni test in BF and ISQ was used. A Pearson correlation was done to investigate the relationship between the BF and DI stability, and $P < 0.05$ was considered statistically significant. The mixed-effects regression to model ISQ ~ BF + time + location + gender, and 95% CIs. If any patient in this study missed a follow-up visit, the author would call the patient to revisit with a new appointment.

## Results

The current study enrolled 128 individuals, and the inclusion criteria were present in 80 individuals needing DIs for the anterior and posterior regions of the jaws in both genders (Fig 2). Descriptive data of patient demographics (number of patients, age, gender, number of implants, implant diameter, and length. The age and gender of participants and implant number, diameter, and length were non significantly different between study groups (Table 1).

Table 2 presents the data on the anterior and posterior BF for the male and female participants after receiving the DI. On average, the male participants exhibited a significantly higher posterior BF compared to the females. Similarly, the average posterior BF in the female patients was significantly lower than in the males at the three visits (Fig 3). These findings suggested that the difference in the OF was based on gender.

The stability at the anterior and posterior DIs for the male and female participants at the three visits is shown in Table 3. The present finding showed that the stability of the anterior DI significantly increased for the male participants compared to the females. Likewise, the average stability of the posterior DIs in the male participants was significantly superior to that of the females, as illustrated in the respective three visits. The result showed that gender differences may have played a potential role in the stability of DIs (Fig 4).

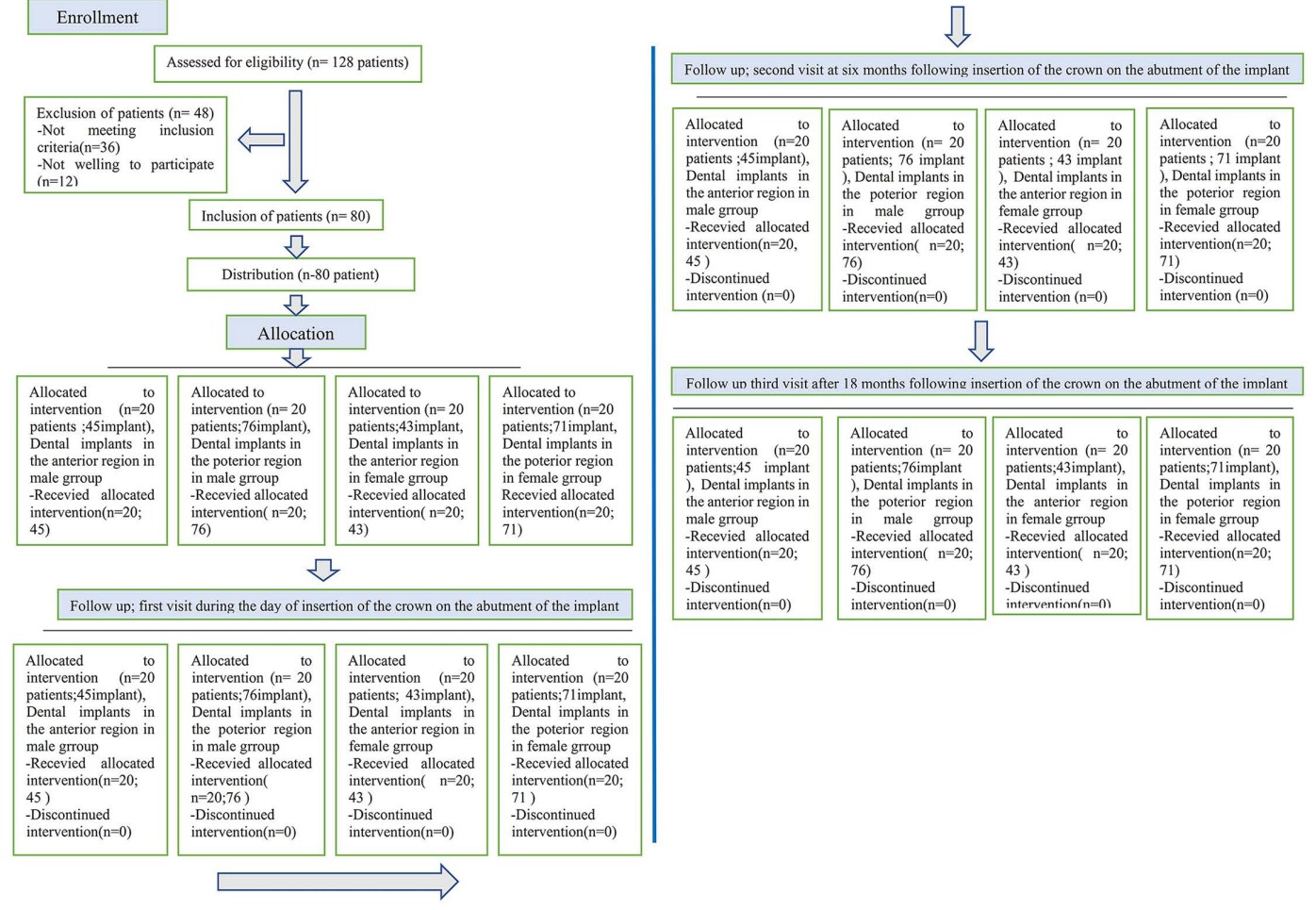

**Fig 2. The CONSORT flow diagram.**

**Table 2. The Anterior & Posterior Bite Forces in Males & Females.**

| NO. of visit | Gender | Anterior Bite Forces(N) mean±SD n=20 | Posterior Bite Forces(N) mean±SD n=20 | Gender | Anterior Bite Forces(N) mean±SD n=20 | Posterior Bite Forces(N) mean±SD n=20 |
|---|---|---|---|---|---|---|
| 1 First | Males | 172.3±10.011 | 32o±4.475 | Females | 152.9±9.573 | 274.2±2.636 |
| 2 Second | | 175.4±8.212 | 324±6.832 | | 156.1±8.325 | 279.2±6.221 |
| 3 Third | | 179.4±6.243 | 337±5.728 | | 159.4±6.436 | 283.5±3.241 |
| Repeated-Measures ANOVA | | 21.202 | 39.831 | | 12.638 | 34.122 |
| p value | | 0.0502 | 0.033 | | 0.0422 | 0.043 |

NO: number; Significant ($p$-values ≤ .05), non-significant ($p$-values > 05), ± SD: standard deviation, N: Newton, 1 st visit: during the day of insertion of the crown on the abutment of the implant (immediately loading), 2nd vist: second visit at six months following insertion, 3 rd visit: third visit after 18 months following insertion.

Results of the two-way ANOVA of all groups for the implant BF and ISQ values between the time, locations of the implants, and gender (Table 4). The BF results showed statistically significant differences in the correlations between time at the 2nd visit (p=0.051) and between the locations of the implants at the 3rd visit (p=0.042), the correlations between time×location at the 2nd visit and 3rd visit (p=0.016; 0.048), respectively. The BF finding revealed a statistically significant difference in the correlations between gender×time at the 2nd visit (p=0.007), and the correlations between gender×location at the 2nd visit and 3rd visit (p=0.032;0.052), respectively. Time×location× gender interaction is significantly correlated with BF.

The ISQ results showed statistically significant differences in the correlations between location at the 2nd visit (p=0.043) and the correlations between lime×location at the 2nd visit and 3rd visit (p=0.011; 0.052), respectively. The ISQ outcomes revealed statistically significant differences in the correlations between gender×time at the 2nd visit (p=0.044), and the correlations between gender×location at the 2nd visit and 3rd visit (p=0.019;0.028), respectively. Time×location× gender interaction is significantly correlated with ISQ.

Results of the sphericity (Mauchly) and handling of violations (Greenhouse-Geisser) were employed (Table 5). The Bite forces results showed statistically non-significant differences in the correlations between time of the 3 visits, the P-value is 0.071, so time has no significant effect on subjects' implant, while locations of the implants have (P_value=0.018). For interaction Time×Location, we can still proceed with the test by using the Greenhouse-Geisser correction and conclude that interaction Time×Location has a significant effect (P_value=0.001).

The Bite forces results showed statistically non-significant differences in the correlations between gender for the three visits, the P-value is 0.093, so gender has no significant effect on subjects' implant, while the interaction of gender×Time has a P-value of 0.024. For interaction Time×Location, we can still proceed with the test by using the Greenhouse-Geisser correction and conclude that the interaction gender×Location has a significant effect (P_value=0.000).

The ISQ results showed statistically non-significant differences in the correlations between time of the three visits, the P-value is 0.063, so time has no significant effect on subjects' implants, while locations of the implants have (P-value 0.059). For interaction Time×Location, we can still proceed with the test by using the Greenhouse-Geisser correction and conclude that interaction Time×Location has a significant effect (P-value=0.000).

The ISQ results showed statistically non-significant differences in the correlations between gender at the three visits, the P-value is 0.001, so gender has no significant effect on subjects' implant, while the interaction of gender×Time has (P-value of 0.007. For interaction Time×Location, we can still proceed with the test by using the Greenhouse-Geisser correction and conclude that the interaction gender×Location has a significant effect (P_value=0.000).

By using the Bonferroni test in intergroup comparisons of mean anterior BF in males, posterior BF in males, anterior BF in females, and posterior BF in females showed significant differences were found among these groups. Intergroup

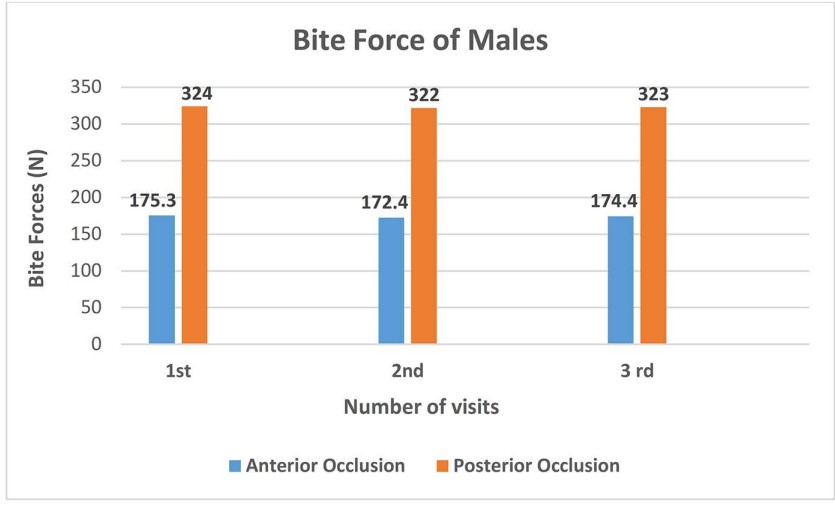

**(a)**

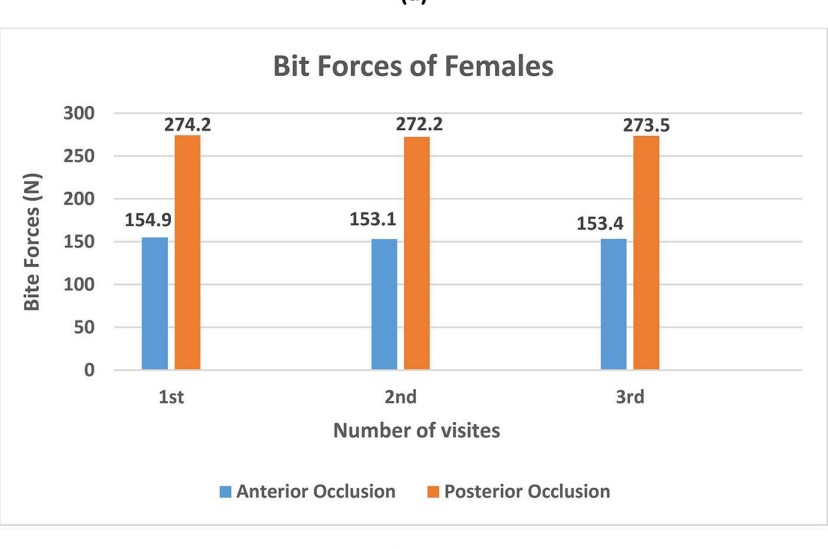

**(b)**

**Fig 3. The mean value differences of anterior bite forces and posterior bite forces in males (a); The mean value differences of anterior bite forces and posterior bite forces in females (b).**

comparisons of mean anterior DI stability in males, posterior DI stability in males, anterior DI stability in females, and posterior DI stability in females showed significant differences among these groups (Table 6).

The data showed that the correlation between the BF and DI stability was significant in the male and female participants (Table 7).

The difference in the regression coefficient (*b*) for BF and ISQ is influenced by time, implant location, and gender. The relationship was observed over time change in three visits, which influenced (BF and ISQ), it was significantly higher BF, ISQ value (3 third visit) when compared (1 first visit, 2nd visit). Such a straightforward association was detected in implant location change in anterior and posterior areas at three visits where effected on (BF and ISQ), where it was significantly higher ISQ value (3 third visit), and BF value (2nd visit) when compared to the 1st visit. The association was illustrated by gender change between males and females at three visits, impact (BF and ISQ), it was significantly greater ISQ value

**Table 3. The Anterior and Posterior Dental Implant Stability in Males and Females.**

| NO. of visit | Gender | Anterior Stability(ISQ) mean±SD n=20 | Posterior Stability(ISQ) mean±SD n=20 | Gender | Anterior Stability(ISQ) mean±SD n=20 | Posterior Stability(ISQ) mean±SD n=20 |
|---|---|---|---|---|---|---|
| 1 First | Males | 74.6±5.35 | 78.9±3.62 | Females | 70.3±4.3 | 75.5±2.11 |
| 2 Second | | 77.8±4.112 | 86.3±3.22 | | 72.4±1.324 | 78.3±2.512 |
| 3 Third | | 79.2±4.861 | 89.6±1.734 | | 74.5±1.453 | 80.1±2.382 |
| Repeated-Measures ANOVA | | 26.023 | 36.122 | | 23.282 | 29.017 |
| p value | | 0.053 | 0.044 | | 0.051 | 0.037 |

NO: number; Significant (p-values ≤ .05), non-significant (p-values > 05), ±SD: standard deviation, ISQ: implant stability quotient, 1 st visit: during the day of insertion of the crown on the abutment of the implant (immediately loading), 2nd vist: second visit at six months following insertion, 3 rd visit: third visit after 18 months following insertion.

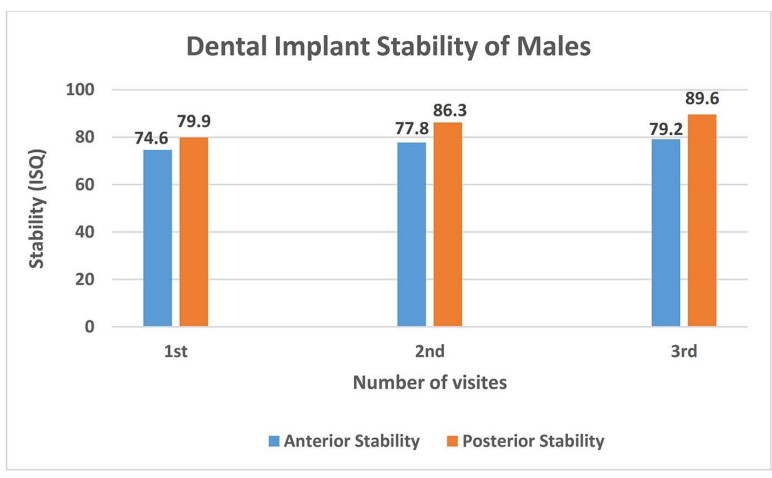

(a)

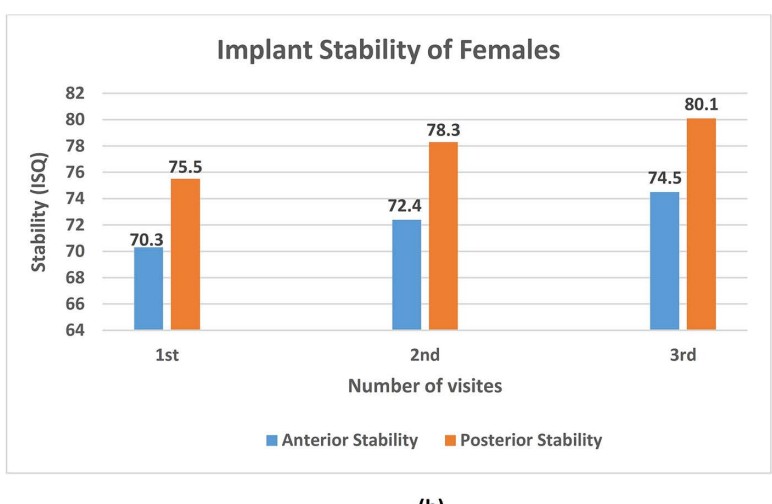

(b)

**Fig 4. The mean value differences of anterior implant stability and posterior implant stability in males (a); The mean value differences of anterior implant stability and posterior implant stability in females (b).**

**Table 4. Results of the two-way and three-way ANOVA of all groups for the bite forces and implant stability values between the time, locations of the implants, and gender.**

| Residual variables | Influence variables | 1 First visit | 2 Second visit | 3 Third visit |
|---|---|---|---|---|
| Bite forces(N) | Time | 0.750 | 0.051* | 0.075 |
| | Locations of implants | 0.657 | 0.534 | 0.042* |
| | Time×Location | 0.134 | 0.016 * | 0.048 * |
| | Gender | 0.119 | 0.052 | 0.157 |
| | Gender×Time | 0.250 | 0.007 * | 0.870 |
| | Gender×Location | 0.720 | 0.032* | 0.052* |
| | Time×Location× Gender | 0.003* | 0.052* | 0.000* |
| StabilityISQ | Time | 0.923 | 0.809 | 0.162 |
| | Locations of implants | 0.185 | 0.043* | 0.137 |
| | Time×Location | 0.977 | 0.011* | 0.051* |
| | Gender | 0.170 | 0.162 | 0.259 |
| | Gender×Time | 0.186 | 0.044 * | 0.199 |
| | Gender×Location | 0.472 | 0.019* | 0.028* |
| | Time×Location× Gender | 0.001* | 0.004* | 0.000* |

\* denotes a significant difference, with p<0.05.

**Table 5. Results of the sphericity (Mauchly), and handling of violations (Greenhouse-Geisser).**

| Residual variables | Influence variables | 1 First visit | 2 Second visit | 3 Third visit | Mauchly | Greenhouse-Geisser |
|---|---|---|---|---|---|---|
| Bite forces(N) | Time | 0.750 | 0.051* | 0.075 | 0.071 | 0.052 |
| | locations of the implants | 0.657 | 0.534 | 0.042* | 0.018* | 0.001 |
| | Time×Location | 0.134 | 0.016 * | 0.042 * | 0.001* | 0.000 |
| | gender | 0.119 | 0.052 | 0.157 | 0.093 | 0.076 |
| | gender×Time | 0.250 | 0.007 * | 0.870 | 0.024* | 0.001* |
| | gender×Location | 0.072 | 0.032* | 0.052* | 0.001* | 0.000* |
| Stability ISQ | Time | 0.923 | 0.809 | 0.162 | 0.063 | 0.099 |
| | locations of the implants | 0.0185 | 0.0436 | 0.0137 | 0.0591* | 0.003* |
| | Time×Location | 0.097 | 0.011* | 0.051* | 0.022* | 0.000* |
| | gender | 0.170 | 0.162 | 0.259 | 0.001* | 0.001* |
| | gender×Time | 0.186 | 0.044 * | 0.191 | 0.007* | 0.000* |
| | gender×Location | 0.472 | 0.019* | 0.028* | 0.004* | 0.000* |

\* denotes a significant difference, with p<0.05.

and BF value (3 third visit) when compared (1 first visit, the 2nd visit. The ratios of the regression coefficients for BF and implant ISQ thus suggest that a unit increase of BF changes ISQ, and BF might prove a more reliable factor for implant stability in differences (time, implant location, and gender) (Table 8).

## Discussion

The bite force (BF) may be a crucial factor in prosthetic design and stability of DIs, particularly in individuals who can produce extremely high occlusal loads [31]. This study found a significant complex interaction of influencing factors (time, location, gender) on the change of BF and ISQ by affecting the process of ossteointegration. The significant changes of

**Table 6. Intergroup Comparisons of Mean Bite Force and Stability in both genders by using the Bonferroni test after adjustment of other factors.**

| Variable | Groups | Groups | Bonferroni test | p value |
|---|---|---|---|---|
| Bte Force | Anterior BF in males | Posterior BF in males | 23.84 | 0.008 |
| | | Anterior BF in females | 11.922 | 0.042 |
| | | Posterior BF in females | 33.42 | 0.055 |
| | Posterior BF in males | Anterior OF in females | 39.21 | 0.057 |
| | | Posterior BF in females | 18.22 | 0.013 |
| | Anterior BF in females | Posterior BF in females | 28.63 | 0.034 |
| Stability | Anterior DIs stability in males | Posterior DIs stability in males | 32.08 | 0.011 |
| | | Anterior DIs stability in females | 21.46 | 0.028 |
| | | Posterior DIs stability in females | 28.36 | 0.046 |
| | Posterior DIs stability in males | Anterior DIs stability in females | 31.68 | 0.003 |
| | | Posterior DIs stability in females | 19.22 | 0.005 |
| | Anterior DIs stability in females | Posterior DIs stability in females | 39.114 | 0.001 |

Significant at ($p < 0.05$); Bonferroni test.

**Table 7. The Correlation between Bte Forces and Dental Implant Stability in Males and Females.**

| Gender | Bite force | Stability | r | p value |
|---|---|---|---|---|
| Males | 243.1 | 75.4 | 0.309 | 0.002* |
| Females | 186.2 | 79.8 | 0.323 | 0.012* |

r: Pearson correlation coefficient; *: Significant ($p$-values ≤ .05), and non-significant ($p$-values > 05).

**Table 8. Differences in regression coefficients (b) for BF and implant ISQ between the Time, Location of Implants, and Gender. b illustrated the mean percentage change in one unit of influencing variables (Time, Location of Implants, and Gender) That is based on the change in BF and implant ISQ. Significant differences in the regression coefficients (b) are denoted by * (p<0.05). The term ratio reflects the quotient of bISQ/ bBF.**

| Variables | | 1 First visit | | 2 Second visit | | 3 Third visit | |
|---|---|---|---|---|---|---|---|
| Influencing variables | Residual varibles | b (%) | 95% CI | b (%) | 95% CI | b (%) | 95% CI |
| Time | BF | 12.4 * | (10.7-13.8) | 16.1* | 4.6 10.2 | 19.8* | 13.5 17.9 |
| | ISQ | 39.3 * | (25.5-56.9) | 49.2 | 66.7 107.6 | 58.4* | 70.2 97.1 |
| | Ratio | 3.1 | | 3.0 | | 2.9 | |
| Location of Implants | BF | 19.4 * | (12.6- 17.6) | 22.8* | 5.5 10.1 | 24.6 | 3.9 5.8 |
| | ISQ | 40.8 * | (79.2- 86.0) | 56.2* | 127.3 83.2 | 66.4* | 44.0 58.5 |
| | Ratio | 2.1 | | 2.4 | | 2.6 | |
| Gender | BF | 16.4 * | 12.6 17.6 | 24.5* | 13.5 17.9 | 30.2* | 14.4 18.4 |
| | ISQ | 62.5* | 79.2 86.0 | 78.3 | 82.2 97.1 | 88.1* | 53.7 102.3 |
| | Ratio | 3.8 | | 3.1 | | 2.9 | |

* denotes a significant difference, with p<0.05.

(time, location, gender) during 18 months influence an increase in BF, and change ISQ with (primary, secondary) stability. The significantly higher mean BF in the posterior regions of males after delivery of dental implant restoration, and the significantly greater mean stability of DIs in the posterior regions during the 18 months following insertion of the crown, within three follow-up visits, could play a potential role in great DIs therapy. This study found a significant association between

BF and DI stability with the impact of time, location of implants, and gender. Therefore, the null hypothesis is that there was no change in BF and ISQ of DIs during 18 months, following insertion of the crown on the abutment of the implant was rejected.

The current study found that the BF in the posterior DIs of males and females was higher than in the anterior DIs. Thus, the BF of the patient may be a concern for a good result in in the DI therapy, especially in the maxillary anterior DI in both genders, since the facial cortical bone may be most susceptible to overload. The bite force (BF) could be a crucial planning factor. The size, quantity, and occlusal design elements of the DI that can effectively withstand the load may be required for a patient who creates an excessive load [32,33]. A high BF capacity may indicate a high risk for a late component fracture [34]. The BF on the posterior regions of males and females is always larger than on the anterior regions. Different occlusion circumstances primarily influence the amplitude and direction of the BF. The DI and the entire mandible underwent noticeably increased stress under occlusion [35]. The DI length, location, an anterior-directed occlusal scheme, splinting, and ridge expansion augmentation may be used to improve osseous support or deflect or reduce the BF [36]. Even with poor anatomical bone characteristics, a patient with a moderate BF may be able to have a satisfactory long-term result [37,38].

The study revealed that there was an increase in the stability of the posterior DIs of the male and female participants compared to the anterior DIs. Hence, to effectively incorporate the DI into the bone, it is first necessary to establish the primary and secondary stability of the DI. The success rate of dental implants (DIs) may be highly dependent on the stability of the patient's DI [39]. However, early failure can happen in a DI for several reasons, even before the restorative component is inserted. Factors that have been positively associated with a higher risk of failure include advanced age, diabetes, cigarette smoking, and lengthier DIs [40].

There are two types of DI stability: primary stability, which involves the mechanical connection of the DI to the bone around its placement, and secondary stability, which involves the tissue reaction to the DI and the subsequent bone remodelling events [41]. The posterior region of the mouth consists of dense trabecular bone with a thick cortical plate compared to the anterior part of the mouth. Therefore, the primary stability is higher in this region of the mouth [42].

The correlation between the OF and DI stability was significant in males and females concerning how the BF initiates bone loss around the DI or how it relates to DI stability. Even after the DI has been joined with the bone, the BF can affect the DI-bone contact and the cells that rebuild the bone in various ways, which can affect whether or not the integration is maintained [43,44].

This study found a significantly complex interaction of factors (time, location, gender) influencing on change of BF and ISQ by affecting the process of osseointegration. The significant changes of (time, location, gender) during 18 months influence an increase in BF, and changeISQ with (primary, secondary) stability. It varies in a dynamic manner as the interface between the bone and the implant matures, and the patient's sex, time, and anatomical location all play a significant role in its context. Accordingly, bite force with influencing variables, BF, frequently rises when patients adjust to implants over time (healing/adaptation), with notable alterations occurring within as little as three months after operation and persisting until osseointegration occurs and strengthens [45]. Bite force is greatly influenced by location across the jaws; forces rise from the anterior to the posterior region (molars have the greatest BF). Optimum biting BF is significantly influenced by gender, with men often using higher forces than women. In addition, the interaction of variables, including the considerable time x location interaction, indicates that BF increases as time passes in different ways at different locations [46]. Time has a different influence on men and women, according to the gender x time interaction. There is a considerable difference in bite force between males and females at different regions due to the gender x location interaction, with males having a more posterior BF. The influence of time varies between males and females, according to the gender x time interaction [47]. The strong interaction between time, location, and gender may indicate that the impact of the position of DI on biting force as time passes differs for males and females.

Time (duration) was estimated to have a significant positive main effect on ISQ; over time, ISQ usually rises as osteous tissue remodelling. The position in the jaw has a major impact because the posterior portion has harder bone than the anterior region, which typically provides superior primary ISQ. Males frequently have more BF, but ISQ varies greatly by sex; yet, some research indicates that females exhibit higher ISQ rises with time. Moreover the interaction of variables, including the considerable time x location interaction, indicates that ISQ improvements as time passes vary depending on the location of the jaw (e.g., more rapid improvement in the posteror area) [48]. The interplay between gender and location may indicate that males and females possess distinct jaw region stability patterns. The gender x time interaction may show that the sexes' progressions in ISQ improvement as time passes vary [28,49]. The most complicated is the three-way interaction between time, location, and gender on ISQ, which illustrates how all three variables interact (e.g., stability increases as time passes in the posterior region differently in males vs. females). A thorough knowledge of the factors involved can be obtained by employing a three-way ANOVA to assess the separate as well as combined impacts of time, site, and sex on implant ISQ.

Differences in the regression coefficient "b" (showing the strength of the relationship) for BF and ISQ can be seen based on time, location of the implant, and gender, as these factors significantly impact the variables that are dependent variables. Bite force associated time, BF on implant-retained crowns, is considerably less than on actual teeth at first, but rises significantly during the preliminary 6 to 18 months' post-implantation as osseointegration improves and patients adjust [50]. The regression coefficient "b" for time would be positive, demonstrating an increase in force during visits, particularly in the initial phases. Bite force associated location, BF, is typically greater in the posterior molar region than in the incisor portion. Thus, the regression coefficient "b" for location would show a stronger association in the posterior regions [51]. Bite force is associated with gender; males often have much greater BF and pressures of biting than females. The regression coefficient for gender would reveal the stronger association with greater BF in males [52].

Dental implant stability is associated with time. At the time of operation, the main stability is mostly mechanical. Lateral stability varies between visits and is a result of physiological osseointegration as time passes.ISQ readings typically fluctuate, occasionally declining at first, later rising once more. And the regression coefficient "b" for time between insertion and exposure is positive, representing a rise in ISQ through the healing period [53]. Dental implant stability is associated with location; the posterior jaw region has an ISQ that is typically higher than the anterior portion. And, as a result of denser bone quality. Dental implant stability is associated with gender; the impact of gender on ISQ results is indicated by a substantial difference in the regression coefficient "b" for ISQ in males and females [54].

Differences in Regression Coefficients "b" ratio of dental implant, ISQ coefficient "b" to bite BF coefficient "b" with change of time is positively increases ISQ with recovery time to positively increases BF with time, especially 6-18 months [55]. The ratio of b ISQ/ b BF with different locations is a higher "b" value, greater initial ISQ for the posterior vs. anterior region of jaw over time, and posterior implants frequently had greater BF values. The ratio of b ISQ/ b BF with gender (male, female), a high ISQ of males was found as a result to have a substantial effect, to a higher BF "b" value for males vs. females [56].

Primary stability has also been demonstrated to be influenced by the diameter, surface features, and length of implants. The present study used dental implants with sizes (4.0*10, 3.5*11, 3.3*10, or 3.8*9) that showed highly primary and secondary stability. More surface area and a stronger mechanical connection to the tissue around it are provided by rough surfaces of implants [57]. Sandblasted implant surfaces facilitate osteogenesis by increasing osteoblast proliferation and cellular metabolism, according to research conducted in vitro [58,59]. Research has demonstrated the presence of surface response and interaction of cells [60]. Compared to implants with a machining surface, those with acid-etched coatings can achieve a much better bone-to-implant contact in areas with low-quality bone [61]. Experimental evidence has demonstrated that in situations when the amount of bone is limited and implants with diameters below 3.0 millimeters offer adequate initial stability [62]. According to Aparicio et al.'s study on RFA procedures, variables such as supracrestal implant length, abutment length, and upper or lower jaw bone density appear to affect RFA values [63]. The dimensions

of implant outcome influence on stability are consistent with the Raz et al. study's findings, which show that the stability evaluations show greater values for longer implants than for shorter ones, and for densely embedded bone as opposed to softer bone [41].

Implants with a severe thread pattern may improve initial stability [27]. The stability values of tapering implants were consistently higher than those of cylinder implants [32]. These changes involve platform switching and microthreads of DIs used in this study. Compared to non-platform switching designs, platform-switching setups have demonstrated efficient stress performance and reduced the possibility of overloading [64]. The maximum von Mises, compressive, and tensile stresses are reduced when oblique forces are applied to a DI with a platform-switching design compared to a traditional design [65]. The palatal side of the platform and the entire implant surface get a redistribution of the pressures that are moved from the compact bone area to the cancellous bone area [20]. The maximal stresses at the cortical region were lower with platform-switching implants than with conventional implants. Implants with platform switching decreased stress by 40% when subjected to oblique loads and 36% when subjected to axial stresses [21].

After osseointegration and throughout the duration of their use, it is acknowledged that all implants exhibit some degree of loss of bone. According to several claims, the addition of microthreads or "retention grooves" to the implant's neck may help to distribute stress and lessen the amount of bone loss that occurs after installation [66]. In practice, the surgical method and the use of platform switching are linked to the preservation of crestal bone [67]. Additionally, it appears that the progressive thread pattern reduces the crestal bone compression process, hence preventing crestal bone loss [67].

The von Mises stress distribution on the DI system demonstrated that the high stresses on the DI resulted from the action of external forces that primarily occurred close to the DI, where it made contact with the abutment [68]. As a result, when the tooth experienced external stress, the neck of the DI was immediately deformed. Hooke's law predicted that a lot of tension would be produced in this region [69]. The alveolar bone of the DI, next to those that were impacted by external forces, was also discovered to be subject to significant stress as a result of deformation [70]. Additionally, it was clear from looking at the tension on the abutment and abutment screw that the significant stress on the abutment originated from its intersection with the DI [71,72]. The high stress on the abutment screw developed at the point where the screw head was attached to the abutment and where the geometric shape of the screw head and screw bent [73,74]. Therefore, excessive stress caused by the BF should be avoided in the design of the abutments and abutment screws. Otherwise, the DI system may be worn down since the patient will be chewing with it for a prolonged period. By analysing the BF, dental experts can better understand the functional strain exerted on the DI system [75]. This will facilitate the choice of suitable DI parts and materials that can withstand the stresses generated during chewing and biting [76,77]. Clinicians can avoid premature mechanical failures such as screw loosening, abutment fracture, or even DI failure by taking into account the patient's BF [78–80]. Understanding the distribution of the BF can also help to achieve occlusal stability, reduce possible issues, and enhance patient comfort [81,82]. The limitation of this study was that patients who smoked and were hypertensive were not included to avoid bias. The small sample size was included in this study, which caused non-significant differences between study groups, and the increase in subject numbers required more time for work and follow-up visits over the period of the study decision. This study was absent of a control group to compare with because the inclusion criteria of the cohort subjects were not specified. The old age patients, bone quality/density, implant [maxilla/mandible], and delayed loading were not recorded in the data collection because these were not contained within the study criteria. The reaction force on the fixed top end of the DI system can serve as a biomechanical reference for future work on DI designs. Further studies of dental implant stability require follow-up visits for 5-10 years to improve the success of implants.

## Conclusion

This study found a significant complex interaction of influencing factors, time, location, and gender on the change of BF and ISQ by effect on the process of osseointegration. The significant changes of time, location, and gender during 18

months influence an increase in BF, and changeISQ with (primary, secondary) stability. The significantly higher mean BF in the posterior regions of males after delivery of dental implant restoration, and the significantly greater mean stability of DIs during the 18 months following insertion of the crown, within three follow-up visits, could play a potential role in great DIs therapy. This study found a significant association between BF and DI stability with the impact of time, location of implants, and gender. The BF was higher on the posterior DIs area in males due to different occlusion circumstances primarily affect the amplitude and direction of the BF. Besides, the result of DI stability revealed that it increased in the posterior regions of males compared to the anterior regions due to denser bone quality. It clarified that the effective incorporation of the DI into the bone was important for the establishment of the primary and secondary stability.

In addition, the relationship between the BF and DI stability is significantly established with the most important factors influencing the change in BF and DI stability. It varies in a dynamic manner as the interface between the bone and the implant matures, and the patient's sex, time, and anatomical location all play a significant role in its context. Moreover, understanding of the BF impact and ensuring the stability, can implement appropriate treatment strategies and best therapy for DIs.

## Supporting information

**S1 File. Data for analysis.** doi: https://doi.org/10.6084/m9.figshare.29369663.v1. The data contains the recording of the bite force of the male and female participants after the insertion of dental implants for the anterior and posterior regions of the jaws. The dental implants' stability was monitored at the anterior and posterior regions of both genders' jaws. Data are available under the terms CC0.
(XLSX)

**S2 File. STROBE statement checklist.** doi: https://doi.org/10.6084/m9.figshare.29364278.v1.
(DOCX)

**S1 Data. Data in figshar.**
(XLSX)

## Acknowledgments

The authors would like to thank Mustansiriyah University/College of Dentistry, Baghdad, Iraq (www.uomustansiriyah.edu.iq), and the University of Al-Ameed/College of Dentistry for their support during work.

## Author contributions

**Conceptualization:** Nawres Bahaa Mohammed.

**Data curation:** Nawres Bahaa Mohammed, Zina Ali Daily.

**Formal analysis:** Nawres Bahaa Mohammed, Athraa Ali Mahmood.

**Funding acquisition:** Nawres Bahaa Mohammed, Zina Ali Daily, Mohammed Rhael Ali, Noor Fathi Kazim.

**Investigation:** Nawres Bahaa Mohammed, Mohammed Rhael Ali.

**Methodology:** Nawres Bahaa Mohammed, Mohammed Rhael Ali, Noor Fathi Kazim.

**Project administration:** Mohammed Rhael Ali.

**Resources:** Zina Ali Daily, Mohammed Rhael Ali, Noor Fathi Kazim.

**Software:** Hashim Mueen Hussein.

**Supervision:** Hashim Mueen Hussein, Athraa Ali Mahmood.

**Validation:** Hashim Mueen Hussein.

**Visualization:** Zina Ali Daily, Noor Fathi Kazim.

**Writing – original draft:** Zina Ali Daily, Noor Fathi Kazim.

**Writing – review & editing:** Zina Ali Daily, Athraa Ali Mahmood.

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
