## [Decision Letter · Decision Letter 0]

6 May 2025

Dear Dr. Hussein,

We look forward to receiving your revised manuscript.

Kind regards,

Antonio Riveiro Rodríguez, PhD

Academic Editor

PLOS ONE

Journal Requirements:

2. In this instance it seems there may be acceptable restrictions in place that prevent the public sharing of your minimal data. However, in line with our goal of ensuring long-term data availability to all interested researchers, PLOS’ Data Policy states that authors cannot be the sole named individuals responsible for ensuring data access (http://journals.plos.org/plosone/s/data-availability#loc-acceptable-data-sharing-methods).

Reviewers' comments:

Reviewer's Responses to Questions

**Comments to the Author**

1. Is the manuscript technically sound, and do the data support the conclusions?

Reviewer #1: No

Reviewer #2: Yes

2. Has the statistical analysis been performed appropriately and rigorously?

Reviewer #1: I Don't Know

Reviewer #2: Yes

3. Have the authors made all data underlying the findings in their manuscript fully available?

Reviewer #1: No

Reviewer #2: Yes

4. Is the manuscript presented in an intelligible fashion and written in standard English?

Reviewer #1: Yes

Reviewer #2: No

Reviewer #1: Author has made a good attempt to study . However the study lacks standardisation to categorically observe the effect of bite force and stability.

The introduction is too long. it should be written around the problem. There should be a problem statement. what was the research question? What were the objectives? Was there any hypothesis?

Methodology: The instrument used can only be used to assess the bite force of posteriors. It is not suitable for use in anteriors since it does not imitate the anterior force direction on Implants.

Consider to use how this instrument was used in the methodology. There is no need to explain the mechanism of action of ostell ISQ instrument. Was the study done on Immediately loaded or immediate placement and loading ? please add clarification.

All the instrument and materials shall be mentioned the commercial name, company of manufacture and country of manufacture in brackets as mentioned for first time.

consider rewriting the methodology

Limitations of the study and further scope for the study need to be mentioned.

conclusion shall not be descriptive . It has to be written in brief and against the objectives of the study

Reviewer #2: Thank you for the interesting manuscript, however, one issue needs to be addressed, which is the type of opposing occlusion, please mention this and include it in the statistical analysis as a group comparison if some patients had natural occlusions and others had artificial occlusion opposing the implants.

Also, please correlate your findings with studied implants dimensions

please clearly explain the impact of using different thread designs in the tested implants.

please have your manuscript edited by a linguistic professional.

**Do you want your identity to be public for this peer review?** For information about this choice, including consent withdrawal, please see our Privacy Policy

Reviewer #1: No

Reviewer #2: No

---

## [Author Response · Author response to Decision Letter 1]

21 Jun 2025

Thank you for your letter and for the reviewers’ comments concerning our manuscript entitled “. The impact of bite force on the long-term success of dental implants”. Those comments are all valuable and very helpful for revising and improving our paper, as well as the important guiding significance to our research. We have studied comments carefully and have made corrections which we hope meet with approval. The revised portion is marked green and yellow in the paper. The main corrections in the paper and the responses to the reviewer’s comments are as follows:

Academic Editor of PLOS ONE: Thank you for providing information to enhance the manuscript about the Journal Requirements

1-Response to comment: Please ensure that your manuscript meets PLOS ONE's style requirements

Response: Thank you for your comment. I checked and modified the manuscript according to PLOS ONE's style.

2. Response to comment: The goal of ensuring long-term data availability to provide all interested researchers

Response: Thank you for pointing that out.. We were adding Data Availability Statement, the data that support the findings of this study are available on request from the corresponding author and emails address, Drdhuhahhaziz@gmail.com, dr.rabeemajeed@gmail.com

Responds to the reviewers’ comments:

Reviewer #1: Thank you for your valuable feedback. I will revise the manuscript to include a clear sentence in green.

1- Response to comment: Author has made a good attempt to study. However, the study lacks standardisation

Response: Thank you for your comment. I clarify that this study has pre-inserted implant measurements, and these readings depend on digital measurements that gave a standardized measurement of bite force to the natural anterior and posterior teeth, but these readings did not show significant differences from the bit force record after primary loading of a crown on abutment of implant and started function.

Moreover, this study has pre-inserted abutment (initial) stability measurements, and these readings were not significantly different from the stability record after primary loading of the crown on the abutment of the implant and started function.

2- Response to comment: The introduction is too long. it should be written around the problem. There should be a problem statement. what was the research question? What were the objectives? Was there any hypothesis?

Response: Thank you for your comment. You're correct that the research question, objectives, and hypothesis were written at the end of the introduction and shortening the introduction too.

The research question is whether the BF impact on stability and longevity of DIs. This study aims to explore the effect of the BF on DIs of different genders, to investigate DI stability in both genders, and to assess the correlation between BF and DI stability. The null hypothesis was that the BF could not have a significant impact on the stability and longevity of DIs

3- Response to comment: Methodology: The instrument used can only be used to assess the bite force of posteriors. It is not suitable for use in anterior since it does not imitate the anterior force direction on Implants

Response: Thank you for pointing that out. I will revise the instrument in Methodology to make it clearer and more explicit with references that the loadstar sensor was used to collect the BF values of posterior and anterior implant areas [28].

4- Response to comment: Consider to use how this instrument was used in the methodology. There is no need to explain the mechanism of action of ostell ISQ instrument. Was the study done on Immediately loaded or immediate placement and loading ? please add clarification.

Response: Thank you for your comment. The method for monitoring implant stability during healing and immediately loading. The mechanism of action of the Ostell ISQ instrument is very important to give information on how it works and obtain measurements.

5-Response to comment : All the instrument and materials shall be mentioned the commercial name, company of manufacture and country of manufacture in brackets as mentioned for first time.

Response: Thank you for your suggestion. The Osstell (Integration Diagnostics AB, Göteborg, Sweden).Loadstar sensor, DI-100U, 16-bit load cell interface, Fremont. California. Straumann, Dental Implant System. USA. Medintika, Dental Implant.Germany.

6-Response to comment: Consider rewriting the methodology.

Response: Thank you for pointing that out. I will revise the i methodology to make it clearer and more explicit. The manuscript methodology was modified according guidelines of the PLOS One journal.

7-Response to comment: Limitations of the study and further scope for the study need to be mentioned.

Response: Thank you for pointing that out. The limitations and further scope were added at the end of the discussion.

The limitation of this study was that patients who smoked and were hypertensive were not included to avoid bias. The small sample size was included in this study, which caused non-significant differences between study groups, and the increase in subject numbers that required more time for work and follow-up visits throughout the study decision. This study was absence of a control group to compare with because the inclusion criteria of the cohort subjects were not met. The reaction force on the fixed top end of the DI system can serve as a biomechanical reference for future work on DI designs. Further studies of dental implant stability that require follow-up visits for 5-10 years to improve the success of implants.

8-Response to comment: Conclusion shall not be descriptive. It has to be written in brief and against the objectives of the study

Response: Thank you for your comment. We modified the conclusion to clarify the result of the study and explained these results with the objectives.

Reviewer #2: Thank you for your important comments. I will revise the manuscript to include a clear sentence in yellow.

1. Response to comment: one issue needs to be addressed, which is the type of opposing occlusion, please mention this and include it in the statistical analysis as a group comparison if some patients had natural occlusions and others had artificial occlusion opposing the implants.

Response: Thank you for pointing this out. It was clarified by adding to the inclusion criteria, they had natural occlusions opposing their implants that needed to be inserted. Moreover, the exclusion criteria included: they had missing permanent teeth, an implant, and were restored with a crown of opposing occlusion.

2. Response to comment: Also, please correlate your findings with studied implants dimensions

Response: Thank you for your suggestion. We correlate the study findings with implant dimensions by adding clear sentences.

Primary stability has also been demonstrated to be influenced by the diameter, surface features, and length of implants. The present study dental implants used with size (4.0*10, 3.5*11, 3.3*10, or 3.8*9) showed high primary and secondary stability. More surface area and a stronger mechanical connection to the tissue around it are provided by rough surfaces of implants [42]. Sandblasted implant surfaces facilitate osteogenesis by increasing osteoblast proliferation and cellular metabolism, according to research conducted in vitro [43,44]. Research has demonstrated the presence of surface response and interaction of cells [45]. Compared to implants with a machining surface, those with acid-etched coatings can achieve a much better bone-to-implant contact in areas with low quality of bone [46]. Experimental evidence has demonstrated that in situations when the amount of bone is limited and implants with diameters below 3.0 millimeters offer adequate initial stability [47]. According to Aparicio et al.'s study on RFA procedures, variables such as supracrestal implant length, abutment length, and upper or lower jaw bone density appear to affect RFA values [48].

The dimensions of implant outcome on stability are consistent with the Raz et al. study's findings, which show that the stability evaluations show greater values for longer implants than for shorter ones, and for densely embedded bone as opposed to softer bone [39].

3. Response to comment: please clearly explain the impact of using different thread designs in the tested implants.

Response: Thank you for your suggestion.

Implants with a severe thread pattern may improve initial stability [27]. The stability values of tapering implants were consistently higher than those of cylinder implants [29]. These changes involve platform switching and microthreads of DIs used in this study. Compared to non-platform switching designs, platform-switching setups have demonstrated efficient stress performance and reduced the possibility of overloading [49]. The maximum von Mises, compressive, and tensile stresses are reduced when oblique forces are applied to a DI with a platform-switching design compared to a traditional design [50]. The palatal side of the platform and the entire implant surface get a redistribution of the pressures that are moved from the compact bone area to the cancellous bone area [20]. The maximal stresses at the cortical region were lower with platform-switching implants than with conventional implants. Implants with platform switching decreased stress by 40% when subjected to oblique loads and 36% when subjected to axial stresses [21].

After osseointegration and throughout the duration of their use, it is acknowledged that all implants exhibit some degree of loss of bone. According to several claims, the addition of microthreads or "retention grooves" to the implant's neck may help to distribute stress and lessen the amount of bone loss that occurs after installation [51]. In practice, the surgical method and the use of platform switching are linked to the preservation of crestal bone [52]. Additionally, it appears that the progressive thread pattern reduces the crestal bone compression process, hence preventing crestal bone loss [52].

4. Response to comment: please have your manuscript edited by a linguistic professional.

Response: Thank you for pointing that out. We send the manuscript for proofreading and have a certificate for it.

Special thanks to you for your good comments.

We tried our best to improve the manuscript and made some changes in the manuscript. These changes will not influence the content and framework of the paper. And here we highlight the changes marked in yellow and green in the revised paper.

We appreciate for Editors and Reviewers’ warm work, and hope that the correction will meet with approval.

Once again, thank you very much for your comments and suggestions.

We look forward to hearing from you. If you have any queries, please don’t hesitate to contact me anytime at the address below.

Thank you and best regards.

Yours sincerely,

---

## [Decision Letter · Decision Letter 1]

18 Aug 2025

Dear Dr. Hussein,

**Please, address all the comments made by the reviewers, in particular, improve the presentation and the statistical analysis.**

We look forward to receiving your revised manuscript.

Kind regards,

Antonio Riveiro Rodríguez, PhD

Academic Editor

PLOS ONE

**Journal Requirements:**

Reviewers' comments:

Reviewer's Responses to Questions

**Comments to the Author**

Reviewer #2: All comments have been addressed

Reviewer #3: (No Response)

2. Is the manuscript technically sound, and do the data support the conclusions?

Reviewer #2: Yes

Reviewer #3: No

3. Has the statistical analysis been performed appropriately and rigorously?

Reviewer #2: Yes

Reviewer #3: No

4. Have the authors made all data underlying the findings in their manuscript fully available?

Reviewer #2: Yes

Reviewer #3: Yes

5. Is the manuscript presented in an intelligible fashion and written in standard English?

Reviewer #2: Yes

Reviewer #3: No

**Reviewer #2:**  Author have responded to all comments and I have no further comments. the authors have adequately addressed your comments raised in a previous round of review and I feel that this manuscript is now acceptable for publication.

**Reviewer #3:**  some commets to make your paper clearer

1.Unclear and incomplete results presentation. The data tables do not define the meaning of multiple numbers reported in each cell (e.g., whether they represent mean ± SD, confidence limits, or another metric), preventing independent interpretation.

2.Figures suggest non-linear patterns, yet only linear fits are superimposed without justification, obscuring true relationships.

3.Inappropriate and insufficient statistical methodology. Normality‐ or variance-homogeneity checks were not reported, so the suitability of parametric tests is unverified.

4.A one-way ANOVA was applied to compare first, second, and third observations, although a repeated-measures ANOVA (or a mixed model) is required for dependent data.

5.Correlations were assessed with simple linear regression despite visual evidence of non-linearity and the absence of diagnostic plots or residual analyses.

6.Mismatch between stated aims and executed analyses. Also not stattic the hyposthes as suggested.

7.The aim explicitly includes gender comparison (“…effect of BF on DIs of different genders…”), yet no direct gender stratification or interaction term appears in the statistical model.

8.Unsupported and overstated conclusions. The conclusion section reiterates some findings (results) and speculates on gender differences without empirical support.

9.Several claims (e.g., superiority of one gender in long-term stability) extend beyond the study’s data and timeframe.

**Do you want your identity to be public for this peer review?** For information about this choice, including consent withdrawal, please see our Privacy Policy

Reviewer #2:**Yes:** MOHAMED AHMED ALKHODARY

Reviewer #3: No

---

## [Author Response · Author response to Decision Letter 2]

31 Aug 2025

Response to the comment of Reviewer 3

1. Response to comment: 1. Unclear and incomplete results presentation. The data tables do not define the meaning of multiple numbers reported in each cell (e.g., whether they represent mean ± SD, confidence limits, or another metric), preventing independent interpretation.

Response: Thank you for your valuable feedback. I will revise the data tables to include a clear sentence in yellow and add the mean ± SD in the tables for further clarification.

This is revised tables:

Table 1. The Anterior and Posterior Occlusal Forces in Males and Females

NO.

of visit Gender Anterior Occlusion

mean ± SD Posterior Occlusion

mean ± SD Gender Anterior Occlusion

mean ± SD Posterior Occlusion mean ± SD

1 st males 175.3

± 10.011 324

± 4.475 females 154.9

± 9.573 274.2

± 2.636

2 nd 172.4

± 8.212 322

± 6.832 153.1

± 8.325 272..2

± 6.221

3 rd 174.4

± 6.243 323

± 5.728 153.4

± 6.436 273..5

± 3.241

Repeated-Measures ANOVA 21.202 39.831 12.638 34.122

p value 0.0502 0.033 0.0422 0.043

Significant (p-values ≤.05), non-significant (p-values > 05), ± SD: standard deviation.

Table 2. The Anterior and Posterior Dental Implant Stability in Males and Females

NO.

of visit Gender Anterior Stability

mean ± SD Posterior Stability

mean ± SD Gender Anterior Stability

mean ± SD Posterior Stability

mean ± SD

1 st Males 75.5

± 4.3 72.3

± 2.11 Females 79.9

± 5.35 74.6

± 3.62

2nd 72.3

± 4.112 72.8

± 2.512 82.3

± 1.324 75.4

± 3.221

3 rd 74.1

± 4.861 73.2

± 2.382 82.6

± 1.453 76.5

± 1.734

Repeated-Measures ANOVA 29.017 26.023 36.122 31.282

p value 0.0532 0.044 0.0518 0.037

Significant (p-values ≤.05), non-significant (p-values > 05), ± SD: standard deviation.

2. Response to comment: 2. Figures suggest non-linear patterns, yet only linear fits are superimposed without justification, obscuring true relationships.

Response: Thank you for your suggestion. I was explaining of mean value levels of Occlusal Force and Dental Implant Stability in both genders in Figures 2 and 3. The scatter plot should be explicitly stated for clarity.

This is revised:

(a)

(b)

Figure 2. The mean value differences of anterior occlusal forces and posterior occlusal forces in males (a); The mean value differences of anterior occlusal forces and posterior occlusal forces in females (b).

(a)

(b)

Figure 3. The mean value differences of anterior implant stability and posterior implant stability in males (a); The mean value differences of anterior implant stability and posterior implant stability in females (b).

3. Response to comment: 3. Inappropriate and insufficient statistical methodology. Normality‐ or variance-homogeneity checks were not reported, so the suitability of parametric tests is unverified.

Response: Thank you for your comment. I was adding detailed reporting of how these analyses were conducted in the revision. I will rewrite the statistical analyses methodology of normality, another test for comparisons between more than two groups, and P < 0.05 was considered statistically significant.

This is revised:

Statistical Analyses

Descriptive statistics through GraphPad® Prism 9.5.1 were used to summarize the patient demographics, BF measurements, and DI stability measurements. The normality of distribution was checked by the Shapiro–Wilk and D’Agostino’s tests. The continuous variables were expressed as the mean standard deviation (±), while the categorical variables were presented as frequencies and percentages. Comparative analyses, such as an analysis of variance (Repeated-Measures ANOVA), were performed to assess differences in the BF and DI stability in males and females at the anterior and posterior areas of the jaws among three postoperative time points. For comparisons between more than two groups, Tukey’s honestly significant difference Post Hoc tests were used. A Pearson correlation was done to investigate the relationship between the BF and DI stability, and P < 0.05 was considered statistically significant. If any patient in this study missed a follow-up visit, the author would call the patient to revisit with a new appointment.

4- Response to comment: 4. A one-way ANOVA was applied to compare first, second, and third observations, although a repeated-measures ANOVA (or a mixed model) is required for dependent data.

Response: Thank you for your question. I was writing a new analysis of these data, depending on repeated-measures ANOVA and p-value in Tables 1 and 2 in revision.

This is revised:

Table 1. The Anterior & Posterior Occlusal Forces in Males & Females

NO.

of visit Gender Anterior Occlusion

mean ± SD Posterior Occlusion

mean ± SD Gender Anterior Occlusion

mean ± SD Posterior Occlusion mean ± SD

1 st males 175.3

± 10.011 324

± 4.475 females 154.9

± 9.573 274.2

± 2.636

2 nd 172.4

± 8.212 322

± 6.832 153.1

± 8.325 272..2

± 6.221

3 rd 174.4

± 6.243 323

± 5.728 153.4

± 6.436 273..5

± 3.241

Repeated-Measures ANOVA 21.202 39.831 12.638 34.122

p value 0.0502 0.033 0.0422 0.043

Significant (p-values ≤.05), non-significant (p-values > 05), ± SD: standard deviation.

Table 2. The Anterior and Posterior Dental Implant Stability in Males and Females

NO.

of visit

Gender Anterior Stability

mean ± SD Posterior Stability

mean ± SD Gender Anterior Stability

mean ± SD Posterior Stability

mean ± SD

1 st Males 75.5

± 4.3 72.3

± 2.11 Females 79.9

± 5.35 74.6

± 3.62

2nd 72.3

± 4.112 72.8

± 2.512 82.3

± 1.324 75.4

± 3.221

3 rd 74.1

± 4.861 73.2

± 2.382 82.6

± 1.453 76.5

± 1.734

Repeated-Measures ANOVA 29.017 26.023 36.122 31.282

p value 0.0532 0.044 0.0518 0.037

Significant (p-values ≤.05), non-significant (p-values > 05), ± SD: standard deviation.

5- Response to comment: 5. Correlations were assessed with simple linear regression despite visual evidence of non-linearity and the absence of diagnostic plots or residual analyses.

Response� Thank you for this suggestion. A Pearson correlation was done to investigate the relationship between the BF and DI stability in both genders Table 4. As well as the relationship was illustrated in a diagnostic scatter plot between the OF and DI stability.

This is revised:

(a)

(b)

(c)

(d)

(e)

(f)

(g)

Figure 4 Scatter plot showing the relationship between Anterior OF in males and Anterior DIs stability in males (a); Scatter plot showing the relationship between Posterior OF in males and Posterior DIs stability in males (b); Scatter plot showing the relationship between Anterior OF in females and Anterior DIs stability in females (c ); Scatter plot showing the relationship between Posterior OF in females and Posterior DIs stability in females (d); Scatter plot showing the relationship between OF in males and DIs stability in males (e); Scatter plot showing the relationship between OF and DIs stability in females (f); and Scatter plot showing the relationship between OF and DIs stability (g).

6- Response to comment: 6. Mismatch between stated aims and executed analyses. Also, not static the hypothesis as suggested.

Response: Thank you for your valuable feedback. I will revise the stated aims and executed analyses to include a clear sentence for further clarification.

This is revised:

This study aims to explore the effect of the BF on DIs of different genders, to investigate DI stability in both genders, and to assess the correlation between BF and DI stability in both genders. The null hypothesis is that the BF could not have a significant impact on the stability of DIs. As well as the relationship was illustrated in a diagnostic scatter plot between the OF and DI stability in both genders.

7- Response to comment: The aim explicitly includes gender comparison (“…effect of BF on DIs of different genders…”), yet no direct gender stratification or interaction term appears in the statistical model.

Response: Thank you for pointing that out. I was adding details of the test for comparisons between more than two groups, Tukey’s honestly significant difference Post Hoc tests were used in revision in Table 3.

Table 3. Intergroup Comparisons of Mean Occlusal Force and Stability in both genders.

Variable Groups Groups Tukey’s HSD Post Hoc test p

Occlusal Force Anterior OF in males Posterior OF in males 22.490 0.000

Anterior OF in females 17.349 0.003

Posterior OF in females 38.205 0.001

Posterior OF in males Anterior OF in females 24.859 0.041

Posterior OF in females 15.715 0.027

Anterior OF in females Posterior OF in females 30.856 0.002

Stability Anterior DIs stability in males Posterior DIs stability in males 29.268 0.002

Anterior DIs stability in females 18.824 0.003

Posterior DIs stability in females 34.179 0.000

Posterior DIs stability in males Anterior DIs stability in females 36.557 0.001

Posterior DIs stability in females 17.911 0.000

Anterior DIs stability in females Posterior DIs stability in females 43.355 0.004

Significant at (p <0.05); Tukey’s HSD Post Hoc test: Tukey (honestly significant differences

8- Response to comment: 8. Unsupported and overstated conclusions. The conclusion section reiterates some findings (results) and speculates on gender differences without empirical support.

Response: Thank you for your question. I revised the conclusion, which comprises a clear sentence for further clarification, depending on the strength of correlation between OF and DIs stability in both genders, and the potential impact in both genders.

This is revised:

Conclusion

The significant correlation between BF and DI stability in both genders plays a noteworthy role in the long-term success of DIs. The significantly lower BF in the anterior regions of females and males may have an impact role in the significantly greater stability of DIs. The OF was lower on the anterior DIs area in both genders due to the use of implants for aesthetic purposes, and less functionally. Besides, the result of DI stability revealed that it increased in the anterior regions of females and males compared to the posterior regions, it clarified that the effective incorporation of the DI into the bone was important for the establishment of the primary and secondary stability.

In addition, the inverse relationship between the OF and DI stability may have been due to the positioning of the posterior DI and the greater masticatory BF, which represented the most important factors influencing the increased OF and decreased DI stability. Moreover, the understanding of the BF impact, performing a thorough OF analysis, can implementing appropriate treatment strategies are essential for ensuring the stability and longevity of DIs.

9- Response to comment: 9. Several claims (e.g., superiority of one gender in long-term stability) extend beyond the study’s data and timeframe.

Response: Thank you for your question. This study found a significant correlation between BF and DI stability in both genders. The significantly lower mean BF in the anterior regions of females and males after delivery of dental implant restoration could play a potential role in the significantly greater mean stability of DIs during the one half year following insertion, with three follow-up visits. The significant correlation between BF and DI stability in both genders plays a noteworthy role in the long-term success of DIs.

Special thanks to you for your good comments.

We tried our best to improve the correction of the manuscript and made some changes in the revision manuscript. These changes will not influence the content and framework of the paper. And here we highlight the changes marked in yellow in the revised paper.

We appreciate for Editors and Reviewers’ warm work earnestly, and hope that the correction will meet with approval.

Once again, thank you very much for your comments and suggestions.

We look forward to hearing from you. If you have any queries, please don’t hesitate to contact me anytime at the address below.

Thank you and best regards.

Yours sincerely,

---

## [Decision Letter · Decision Letter 2]

30 Oct 2025

Dear Dr. Hussein,

We look forward to receiving your revised manuscript.

Kind regards,

Antonio Riveiro Rodríguez, PhD

Academic Editor

PLOS ONE

Journal Requirements:

Additional Editor Comments:

Please, address all the comments from reviewers.

Reviewer's Responses to Questions

**Comments to the Author**

Reviewer #4: All comments have been addressed

2. Is the manuscript technically sound, and do the data support the conclusions?

Reviewer #4: No

3. Has the statistical analysis been performed appropriately and rigorously?

Reviewer #4: No

4. Have the authors made all data underlying the findings in their manuscript fully available?

Reviewer #4: Yes

5. Is the manuscript presented in an intelligible fashion and written in standard English?

Reviewer #4: No

Reviewer #4: Overall appraisal

The revised manuscript addresses an important clinical question — whether patients’ bite force (BF) influences dental implant (DI) stability — using a prospective cohort design with repeated BF and ISQ measurements. The topic is relevant to implant planning and prosthetic design. However, major methodological, statistical, and reporting problems limit confidence in the presented conclusions. The manuscript requires focused revision before it is acceptable for publication.

---

Major strengths

- Clear, clinically relevant question with direct implications for implant loading and prosthetic planning.

- Prospective cohort design with repeated measurements (three time points) and use of objective tools: Loadstar sensor for force and Osstell ISQ for stability.

- Inclusion of both anterior and posterior implant sites and sex-stratified reporting.

- Availability of underlying data and STROBE checklist as supporting files.

Major weaknesses that must be fixed

1. Study population and inclusion/exclusion inconsistency

- Textual contradictions about exclusion criteria (e.g., “Patients with … missing permanent teeth” appears in exclusion list but is central to implant patients). Inclusion/exclusion must be unambiguous and justified.

2. Sample size and group structure

- Power analysis stated 18 per group but no clear definition of the groups (4 groups mentioned). Provide exact group definitions, how 80 participants map to groups, and the assumptions used in the G*Power calculation (effect size, SD, correlation for repeated measures).

3. Statistical methods and reporting

- Normality and homoscedasticity results are asserted but diagnostic outputs and decisions (parametric vs nonparametric) are not shown.

- Choice and implementation of repeated-measures models are unclear: reported ANOVA tables look inconsistent with repeated-measures design, no within-subjects factor structure or sphericity checks (Mauchly) are reported, no correction (Greenhouse-Geisser) if sphericity violated.

- Multiple testing and post hoc strategy are unclear; interaction tests (time × sex × site) are absent.

- Correlation and regression: Pearson r reported despite apparent nonlinearity and use of aggregated values (means) rather than within-subject paired BF–ISQ observations; regression diagnostics and model fit not provided.

4. Outcome definitions and timing

- “Success” described qualitatively but no objective failure events, thresholds, or censoring are reported; ISQ change is used as a surrogate but this must be justified and pre-specified.

- Timing description ambiguous: “second visit at six months” and “third visit after one-half year” — clarify exact intervals (e.g., baseline = day of crown insertion; 6 months; 12 months).

5. Data presentation problems

- Tables contain inconsistent entries, typographical errors and unclear numeric formats (e.g., “272 .. 2”, multiple stray symbols, inconsistent ± placement), making independent verification impossible.

- Figures include linear fits where scatter suggests nonlinearity; many axis labels and units missing and figure quality is poor.

6. Interpretation overreach

- Authors make causal-sounding statements and broad claims about long-term implant success based on short (1 year) observational data and correlational analyses.

- Gender statements are speculative without interaction tests or adjustment for confounders.

7. Confounding and covariates

- Key confounders (age as continuous, bone quality/density, implant length/diameter, implant location [maxilla/mandible], surgical technique, immediate vs delayed loading, opposing dentition, parafunction/bruxism, medication, periodontal status) are either not reported or not adjusted for in analyses.

8. Ethics and data availability statements

- Ethics approval and consent are claimed but supporting documentation (ethics ID is given) should be consistent with institutional format; the data availability statement needs to specify how readers access the dataset and what identifiers are included/removed to preserve anonymity.

---

Required edits (ordered by priority)

1. Methods — participants

- Clarify inclusion/exclusion precisely and remove contradictory phrasing.

- Provide a flow diagram (enrolment → allocation → follow-up → analysis) with numbers lost to follow‑up and reasons.

2. Methods — sample size

- Report the exact G*Power inputs (effect size f or d, α, power, correlation among repeated measures, number of groups/time points) and how 80 participants were chosen.

3. Methods — statistical analysis (revise and expand)

- State data structure: subjects measured at 3 time points; BF and ISQ measured at anterior/posterior sites; sex as between-subjects factor.

- Use an appropriate repeated-measures model: mixed-effects model (preferred) or two-way repeated-measures ANOVA including time (within), site (within) and sex (between), and their interactions.

- Report checks: normality (Shapiro-Wilk) on residuals, homogeneity, sphericity (Mauchly), and handling of violations (Greenhouse-Geisser or mixed model). Show test statistics and p-values.

- For correlations use within-subject paired analyses (e.g., compute BF–ISQ correlation across measurements per subject and use mixed-effects regression to model ISQ ~ BF + time + site + sex + random intercept(subject)). Report fixed effects, 95% CIs, p-values, and model diagnostics (residuals, influence).

- Correct for multiple comparisons or justify not doing so.

4. Reporting of results

- Clean all tables: each cell must show mean ± SD (or median [IQR] if nonparametric). Remove typographical artifacts and ensure consistent units (N for force, ISQ units for stability).

- Report sample sizes for each stratum/time point (n per cell).

- Report effect sizes (partial eta-squared for ANOVA; regression coefficients with 95% CI; r and adjusted R^2 for correlations/regressions).

- Replace aggregated scatter/regression plots using subject-level paired points and overlay appropriate model fits; include residual/diagnostic plots in supplement.

5. Confounder adjustment

- Include implant-level and patient-level covariates in models: implant length/diameter, jaw (maxilla/mandible), bone quality rating (if available), immediate vs delayed loading, opposing dentition, age, and parafunctional habits. Present adjusted analyses and compare to unadjusted.

6. Hypothesis framing and interpretation

- Restate primary and secondary hypotheses, predefined primary outcome (e.g., change in ISQ at 12 months), and avoid causal language; discuss directionality and limits of observational data.

- Tone down claims about long-term success unless long-term (≥5 years) data exist.

7. Figures

- Improve figure clarity: label axes including units, show individual subject points, plot repeated-measures trajectories where helpful, and include legend and sample sizes.

- Replace simple linear fits with fitted mixed-model predictions if modeling repeated measures.

8. Tables of implants and procedural details

- Provide a table summarizing implant characteristics (brand, diameter, length, surface, site distribution) and surgical/prosthetic protocol (immediate loading protocol details, abutment type, whether platform switching used).

9. Data and ethics

- Confirm anonymization procedures for shared data and provide clear instructions for accessing supporting files; ensure consent wording matches data sharing.

- Include ethics committee name, reference number, and date of approval as per journal requirements.

10. Language and editing

- Correct typographical errors and awkward phrasing throughout; standardize use of terms: bite force (BF), occlusal force (OF) — choose one term and use consistently.

---

Statistical issues explained and required changes

- Model selection: Repeated observations per subject violate independence; simple one-way ANOVA or separate ANOVAs per group are inappropriate. Use linear mixed-effects models with random intercept (subject) and, if warranted, random slope (time) to account for within-subject correlation and unequal spacing, and include interaction terms (time × sex, time × site, BF × site).

- Sphericity: For ANOVA approaches report Mauchly’s test and apply Greenhouse-Geisser correction when violated.

- Correlation vs regression: Pearson r on aggregated mean values is misleading. Use subject-level paired BF–ISQ data and mixed regression to quantify association while adjusting for covariates. Report regression coefficient (β) interpreted as ISQ change per unit N BF, with 95% CI.

- Nonlinearity and heteroscedasticity: Inspect scatterplots and residuals; consider transformation or generalized additive models if relationship is non-linear. Report diagnostic plots (residual vs fitted, QQ-plot).

- Multiple testing: Many comparisons presented (sex, site, time, post hoc) — control false discovery (e.g., Bonferroni or report adjusted p-values) or focus on prespecified primary comparison.

---

Ethics, consent, data sharing — required clarifications

- Confirm ethics committee approval number and date; include statement that protocol was prospectively registered if applicable.

- Consent: clarify whether consent included data sharing and the form of anonymization for deposited data.

- Data availability: provide the exact repository link and DOI (supporting files are referenced but specify where and how to access raw data and code to reproduce analyses).

---

My comments on this paper ordered according to the manuscript sections:

1. Title and abstract

- Abstract must mirror revised primary outcome and analytic approach (state adjusted analyses and main effect estimates with CIs). Avoid causal phrasing (“impact on long-term success”) unless supported by study duration.

2. Introduction

- Tighten literature synthesis and explicitly identify the primary hypothesis and primary outcome measure.

3. Methods — participants

- Clarify exact inclusion/exclusion items and present a CONSORT-style (STROBE-adapted) flow diagram.

- Provide rationale for age range and reasons for excluding smokers/systemic disease (possible selection bias).

4. Methods — implants and procedures

- Give per-implant details: number of implants per patient, jaw (maxilla/mandible), tooth position, implant brand/size distribution, abutment type, immediate loading protocol specifics. State whether ISQ measured before and after prosthetic insertion.

5. Methods — BF measurement

- Describe Loadstar calibration, sensor placement protocol (how measured for anterior vs posterior), number of repeated bites averaged, and units (N). Report measurement reliability (intra-operator / test-retest) if available.

6. Methods — stability measurement

- Describe ISQ measurement protocol (Smartpeg type, measurement directions, operator blinding) and handling of multiple ISQ readings per implant.

7. Statistical analysis

- Replace current description with explicit mixed-effects model plan: formula, fixed and random effects, covariates, model selection, diagnostics, and multiple testing control. Provide software and package versions used. Report how missing data were handled (e.g., mixed models assume MAR; list any imputation).

8. Results — descriptive

- Provide baseline table with patient characteristics by sex and by implant site; include implant-level counts and per-timepoint sample sizes.

- Clean up Tables 1–3: ensure all values are mean ± SD with units, n per cell, and clear p-values and test statistics.

9. Results — main analyses

- Present adjusted mixed-model estimates (effect of BF on ISQ) with β, 95% CI, p-value. Provide interaction tests (BF × sex, BF × site). If interaction non-significant, present pooled estimates and exploratory stratified analyses.

10. Results — figures

- Provide subject-level scatterplots (BF vs ISQ) with model-predicted lines and 95% CI; include residual diagnostics in supplement.

11. Discussion

- Restructure: start with main findings (with adjusted estimates), compare to prior studies, discuss biological plausibility, limitations (selection bias, short follow-up, residual confounding, measurement error), and implications.

- Remove causal wording; state that BF is associated with ISQ changes and that further longer-term and interventional studies are needed.

12. Limitations

- Expand: explain possible selection bias from excluding smokers/systemic disease, limited follow-up for “long-term” claims, and lack of randomization.

13. Conclusion

- Rephrase to reflect association rather than causation; avoid overreaching on “long-term success”.

14. Supplementary material

- Provide cleaned dataset and analysis script (R, SAS, SPSS) or at least provide sufficient summary statistics to reproduce results.

---

Overall Suggestion:

- Major revision required. The manuscript addresses an interesting question and has useful primary data, but the issues in methods, statistical analyses, data presentation, and interpretation are substantial and must be corrected before re-review. Recommend resubmission only after the authors implement the required methodological/statistical revisions and provide cleaned tables, improved figures, and appropriately adjusted analyses.

**Do you want your identity to be public for this peer review?** For information about this choice, including consent withdrawal, please see our Privacy Policy

Reviewer #4: No

---

## [Author Response · Author response to Decision Letter 3]

12 Dec 2025

Dear Editors and Reviewers:

Thank you for your letter and for the reviewers’ comments concerning our manuscript entitled “The impact of bite force on the stability of dental implants”. Those comments are all valuable and very helpful for revising and improving our paper, as well as the important guiding significance for our research. We have studied comments carefully and have made corrections which we hope will meet with approval. The revised portion is marked green in the paper.

Response to the comment of Reviewer 4

1. Response to comment: 1. Methods — participants

- Clarify inclusion/exclusion precisely and remove contradictory phrasing.

- Provide a flow diagram (enrolment → allocation → follow-up → analysis) with numbers lost to follow‑up and reasons.

Response: Thank you for your valuable feedback. I will revise the inclusion/exclusion precisely and remove contradictory phrasing to include a clear sentence in green and add a flow diagram.

This is revised

The inclusion criteria were male and female patients aged 25-40 years, who had sufficient bone volume for DI placements instead missing number of natural teeth and had natural occlusions, opposed natural teeth should be presented to implants that needed to be inserted, understood and signed an informed consent form, and were followed up at postoperative visits. Patients with a history of systemic diseases affecting bone metabolism, periodontitis, cigarette smoking, or drinking alcohol were excluded from the study. Moreover, the exclusion criteria included: they had missing permanent teeth which replaced by an implant, and a restoration with a crown of opposing, delayed loading, and parafunctional habits.

2. Response to comment: 2. Methods — sample size

- Report the exact G*Power inputs (effect size f for d, α, power, correlation among repeated measures, number of groups/time points) and how 80 participants were chosen.

Response: Thank you for your suggestion. I was explaining sample size calculation by using G*Power inputs (effect size f, α, power, correlation among repeated measures, number of groups/time points) and how 80 participants were chosen. It depended on the pilot study resulting of 12 patients with DIs placed, three in each group (3 patients placed DIs for anterior regions of male, 3 patients placed DIs for posterior regions of male, 3 patients placed DIs for anterior regions of female and 3 patients placed DIs for posterior regions of female) that BF and ISQ measured during three visits interval (first visit during the day of insertion of the crown on the abutment of the implant (immediately loading), second visit at six months following insertion, and third visit after 18 months following insertion. Then these data were analyzed and resulted in a significant effect size of F at 0.4, and four groups. The total number of participants was around 80 to avoid patient dropouts, and these were divided into four groups.

This is revised

Sample Size Power Analysis

The sample size, calculated by G power, comprised 18 individuals in each group at a power of 80, an α probability of 0.05 that it depended on the pilot study resulting of 12 patients with DIs placed, three in each group (3 patients placed DIs for anterior regions of male, 3 patients placed DIs for posterior regions of male, 3 patients placed DIs for anterior regions of female and 3 patients placed DIs for posterior regions of female) that BF and ISQ measured during three visits interval (first visit during the day of insertion of the crown on the abutment of the implant (immediately loading), second visit at six months following insertion, and third visit after 18 months following insertion. Then these data were analysed and resulted in a significant effect size of F at 0.4, and four groups. The total number of participants was around 80 to avoid patient dropouts, and these were divided into four groups. 80 individuals of both genders who had lost some teeth and needed DIs for the anterior and posterior regions of the jaws.

3. Response to comment. Methods — statistical analysis (revise and expand)

A- State data structure: subjects measured at 3 time points; BF and ISQ measured at anterior/posterior sites; sex as a between-subjects factor.

B- Use an appropriate repeated-measures model: mixed-effects model (preferred) or two-way repeated-measures ANOVA including time (within), site (within), and sex (between), and their interactions.

C- Report checks: normality (Shapiro-Wilk) on residuals, homogeneity, sphericity (Mauchly), and handling of violations (Greenhouse-Geisser or mixed model). Show test statistics and p-values.

D- For correlations, use within-subject paired analyses (e.g., compute BF–ISQ correlation across measurements per subject and use mixed-effects regression to model ISQ ~ BF + time + site + sex + random intercept(subject)). Report 95% CIs, p-values, and model diagnostics (residuals, influence).

E- Correct for multiple comparisons or justify not doing so.

Response: Thank you for your comment.

A-I was adding the state data structure in the data collection part of the article.

B-I was adding detailed reporting of repeated-measures model in two-way repeated-measures ANOVA, including time (within), site (within), and sex (between), and their interactions. These analyses were conducted in the revision of Table 4.

C- I was explaining how to check the normality (Shapiro-Wilk) on residuals and homogeneity. The sphericity (Mauchly) and handling of violations (Greenhouse-Geisser on mixed model) were analysed in Table 5.

D-I was added to table 8 using mixed-effects regression to model ISQ ~ BF + time + site + sex + random intercept (subject) and 95% CIs, p-values

E – I corrected the multiple comparisons with the Bonferroni test in bite force and stability after justifying other factors.

This is revised

A-Data Collection

State data structure, subjects measured at 3 time points; BF and ISQ measured at anterior/posterior sites; sex as a between-subjects factor. Relevant clinical information, such as the patient’s characteristics after placement with a DI by a specialist, medical background, X-ray pictures, and scans of the inside of the mouth, was obtained from the computerised medical records and documents of the patient. A Loadstar™ sensor was used to collect the BF values of posterior and anterior implant areas [28]. This sensor offers various force-measuring solutions with updated data rates of up to 50 KHz, making it suitable for applications that require the BF to be noted soon after the operation and during consultation intervals. Regular monitoring of the BF and DI stability was crucial in the post-implantation phase. Loadstar™ sensor records were utilised to assess the OF and identify any abnormalities or imbalances. While the Osstell Implant Stability Quotient (ISQ) technology was used to evaluate the DI stability by examining the resonance frequency of the DI. The patients were also educated on proper oral hygiene practices, including avoiding excessive BF on the DI, maintaining regular dental visits, and promptly addressing any signs of discomfort or changes in the bite. The data availability was presented in supporting information files.

B- Table 4. Results of the two-way ANOVA of all groups for the implant occlusal forces and stability values between the time, locations of the implants, and gender.

Value 1st 2nd 3rd

Bite forces(N) Time 0.750 0.051* 0.075

locations of the implants 0.657 0.534 0.042*

Time × Location 0.134 0.016 * 0.042 *

gender 0.119 0.052 0.157

gender × Time 0.250 0.007 * 0.870

gender × Location 0.720 0.321 0.520

stabilityISQ Time 0.923 0.809 0.162

locations of the implants 0.185 0.436 0.137

Time × Location 0.977 0.811 0.516

gender 0.170 0.162 0.259

gender × Time 0.186 0.044 * 0.199

gender × Location 0.472 0.199 0.289

* denotes a significant difference, with p < 0.05.

C-. The Shapiro-Wilk test was used to detect whether the quantitative data were normally distributed. The normality of distribution (parametric)was significantly result by used the Shapiro–Wilk. The parametric data was analyzed.

Table 5. Results of the sphericity (Mauchly), and handling of violations (Greenhouse-Geisser

Table 5. Results of the sphericity (Mauchly), and handling of violations (Greenhouse-Geisser

Residual variables Influence variables 1 First visit 2 Second visit 3 Third visit Mauchly Greenhouse-Geisser

Bite forces(N) Time 0.750 0.051* 0.075 0.071 0.052

locations of the implants 0.657 0.534 0.042* 0.018* 0.001

Time × Location 0.134 0.016 * 0.042 * 0.001* 0.000

gender 0.119 0.052 0.157 0.093 0.076

gender × Time 0.250 0.007 * 0.870 0.024* 0.001*

gender × Location 0.072 0.032* 0.052* 0.001* 0.000*

Stability ISQ Time 0.923 0.809 0.162 0.063 0.099

locations of the implants 0.0185 0.0436 0.0137 0.0591* 0.003*

Time × Location 0.097 0.011* 0.051* 0.022* 0.000*

gender 0.170 0.162 0.259 0.001* 0.001*

gender × Time 0.186 0.044 * 0.191 0.007* 0.000*

gender × Location 0.472 0.019* 0.028* 0.004* 0.000*

* denotes a significant difference, with p < 0.05.

D- Table 8. Differences in regression coefficients (b) for BF and implant ISQ between the Time, Location of Implants, and Gender. b illustrated the mean percentage change in one unit of influencing variables (Time, Location of Implants, and Gender) based on the change in BF and implant ISQ. Significant differences in the regression coefficients (b) are denoted by * (p < 0.05). The term ratio reflects the quotient of bISQ / bBF.

Variables 1 First visit 2 Second visit 3 Third visit

Influencing variables Residual

varibles b (%) 95% CI b (%) 95% CI b (%) 95% CI

Time BF 12.4 * (10.7-13.8) 16.1* 4.6 10.2 19.8* 13.5 17.9

ISQ 39.3 * (25.5-56.9) 49.2 66.7 107.6 58.4* 70.2 97.1

Ratio 3.1 3.0 2.9

Location of Implants BF 19.4 * (12.6- 17.6) 22.8* 5.5 10.1 24.6 3.9 5.8

ISQ 40.8 * (79.2- 86.0) 56.2* 127.3 83.2 66.4* 44.0 58.5

Ratio 2.1 2.4 2.6

Gender BF 16.4 * 12.6 17.6 24.5* 13.5 17.9 30.2* 14.4 18.4

ISQ 62.5* 79.2 86.0 78.3 82.2 97.1 88.1* 53.7 102.3

Ratio 3.8 3.1 2.9

* denotes a significant difference, with p < 0.05.

E- Table 6. Intergroup Comparisons of Mean Occlusal Force and Stability in both genders. the Means Comparison tab, select the Bonferroni test for bite force and stability

Variable Groups Groups Bonferroni test p value

Bte Force Anterior BF in males Posterior BF in males 23.84 0.008

Anterior BF in females 11.922 0.042

Posterior BF in females 33.42 0.055

Posterior BF in males Anterior OF in females 39.21 0.057

Posterior BF in females 18.22 0.013

Anterior BF in females Posterior BF in females 28.63 0.034

Stability Anterior DIs stability in males Posterior DIs stability in males 32.08 0.011

Anterior DIs stability in females 21.46 0.028

Posterior DIs stability in females 28.36 0.046

Posterior DIs stability in males Anterior DIs stability in females 31.68 0.003

Posterior DIs stability in females 19.22 0.005

Anterior DIs stability in females Posterior DIs stability in females 39.114 0.001

Significant at (p <0.05); Bonferroni test.

4- Response to comment:4- Reporting of results

A- Clean all tables: each cell must show mean ± SD (or median [IQR] if nonparametric). Remove typographical artifacts and ensure consistent units (N for force, ISQ units for stability).

B- Report sample sizes for each stratum/time point (n per cell).

C- Report effect sizes (ANOVA; regression coefficients with 95% CI; r and adjusted R^2 for correlations/regressions).

Response: Thank you for your question.

A-I was the clear mean ± SD of parametric data in cells of Table 2,3. I added the consistent units (N for force, ISQ units for stability).

B-I was reporting the sample sizes for each stratum/time point (n per cell) in table 2,3.

C- I was writing a new analysis of these data, depending on ANOVA, regression coefficients with 95% CI; r and adjusted R^2 for correlations/regressions.

5- Response to comment: 5. Confounder adjustment: Include implant-level and patient-level covariates in models: implant length/diameter, jaw (maxilla/mandible), bone quality rating (if available), immediate vs delayed loading, opposing dentition, age, and parafunctional habits.

Response� Thank you for this suggestion. I was clarifying the details of implant-level and patient-level covariates in models in Table 1.

This is revised

Table 1. Demographic data.

DIs for the anterior regions of males DIs for the posterior regions of males DIs for the anterior regions of females DIs for the posterior regions of females p value

Number of patients (total) 20 20 20 20 0.11

Age 35 37 36 34 0.18

Gender 20 20 20 20 0.21

Number of implants (total) 45 76 43 71 0.79

Implant diameter

3.3 mm 8 10 7 12 0.122

3.5 mm 19 22 23 21 0.084

3.8 mm 14 24 11 30 0.056

4 mm 4 20 2 8 0.077

Length

9 mm 13 49 15 42 0.095

10 mm 25 22 19 23 0.082

11 mm 7 5 9 6 0.112

Significant (p-values ≤.05), non-significant (p-values > 05

6- Response to comment: 6. Hypothesis framing and interpretation

- Restate primary and secondary hypotheses, predefined primary outcome (e.g., change in ISQ at 12 months), and avoid causal language; discuss directionality and limits of observational data.

- Tone down claims about long-term success unless long-term (≥5 years) data exist.

Response� Thank you for this suggestion. I was corrected primary outcome, secondary outcome, and hypotheses.

This is revised

The hypothesis is that there was a change in BF and ISQ of DIs during 18 months, followed insertion of the crown on the abutment of the implant. The null hypothesis is that there was no change in BF and ISQ of DIs during 18 months, following the insertion of the crown on the abutment of the implant.

Outcome Measures

The main objective of the study was to determine the relationship between the BF and the stability of DIs during the 18 months that followed. The secondary outcomes included influencing time, gender, and the DI location in the jaws on the BF and the DI stability.

7- Response to comment: 7. Figures

- Improve figure clarity

Response: Thank you for your valuable feedback. I added Figure 2. The CONSORT 2010 flow diagram, and clear other figures.

8- Response to comment: 8. Tables of implants and procedural details

Provide a table summarizing implant characteristics (brand, diameter, length, surface, site distribution) and surgical/prosthetic protocol (immediate loading protocol details, abutment type, whether platform switching was used).

Response� Thank you for this suggestion. I added Table 1 summarizing implant characteristics (diameter, length, and number of implants in each size and group). Further surgical procedure (immediate loading protocol details.

This is revised

Table 1. Demographic data.

DIs for the anterior regions of males DIs for the posterior regions of males DIs for the anterior regions of females DIs for the posterior regions of females p value

Number of patients (total) 20 20 20 20 0.11

Age 35 37 36 34 0.18

Gender 20 20 20 20 0.21

Number of implants (total) 45 76 43 71 0.79

Implant diameter

3.3 mm 8 10 7 12 0.122

3.5 mm 19 22 23 21 0.084

3.8 mm 14 24 11 30 0.056

4 mm 4 20 2 8 0.077

Length

9 mm 13 49 15 42 0.095

10 mm 25 22 19 23 0.082

11 mm 7 5 9 6 0.112

Surgical Procedure

The implants distributed were placed in the anterior and posterior regions of the jaws (Table 1). Following dental cone-beam computed tomography (CBCT), the quantity and quality of bone were assessed, and a stent was created for implantation at the proper location for each patient. No bone augmentation treatment was used throughout the implant placement process. As instructed by the manufacturer, the location was drilled using a point Lindemann drill first, then surgical drills. A skilled researcher meticulously drilled every dental implant bed at a consistent length and angles in order to produce comparable insertion torque values (ITV) of about 35 Ncm amongst the implants. A drilling machine specifically made for implant surgery was used for measuring ITV as much as 35 Ncm at around 20 rpm and 8 Hz. The drill unit's handpiece was the only tool used to put each implant. Following surgery, CBCT was utilized to

---

## [Decision Letter · Decision Letter 3]

21 Dec 2025

The impact of bite force on the stability of dental implants

PONE-D-25-01820R3

Dear Dr. Hussein,

We’re pleased to inform you that your manuscript has been judged scientifically suitable for publication and will be formally accepted for publication once it meets all outstanding technical requirements.

Kind regards,

Antonio Riveiro Rodríguez, PhD

Academic Editor

PLOS One

Additional Editor Comments (optional):

Reviewers' comments:

Reviewer's Responses to Questions

**Comments to the Author**

Reviewer #4: All comments have been addressed

2. Is the manuscript technically sound, and do the data support the conclusions?

Reviewer #4: Yes

3. Has the statistical analysis been performed appropriately and rigorously?

Reviewer #4: Yes

4. Have the authors made all data underlying the findings in their manuscript fully available?

Reviewer #4: Yes

5. Is the manuscript presented in an intelligible fashion and written in standard English?

Reviewer #4: Yes

Reviewer #4: Thanks for addressing all my comments.

The paper is now suitable for publication.

The authors have made a significant effort to respond to all queries.

**Do you want your identity to be public for this peer review?** For information about this choice, including consent withdrawal, please see our Privacy Policy

Reviewer #4: No

---

## [Editor Report · Acceptance letter]

PONE-D-25-01820R3

PLOS One

Dear Dr. Hussein,

I'm pleased to inform you that your manuscript has been deemed suitable for publication in PLOS One. Congratulations! Your manuscript is now being handed over to our production team.

Kind regards,

on behalf of

Dr. Antonio Riveiro Rodríguez

Academic Editor

PLOS One